# A variational expectation-maximization framework for balanced multi-scale learning of protein and drug interactions

Jiahua Rao [1], Jiancong Xie[1], Qianmu Yuan [1], Deqin Liu[1], Zhen Wang[1], Yutong Lu[1] ✉, Shuangjia Zheng [2] ✉ & Yuedong Yang [1,3,4] ✉

Protein functions are characterized by interactions with proteins, drugs, and other biomolecules. Understanding these interactions is essential for deciphering the molecular mechanisms underlying biological processes and developing new therapeutic strategies. Current computational methods mostly predict interactions based on either molecular network or structural information, without integrating them within a unified multi-scale framework. While a few multi-view learning methods are devoted to fusing the multi-scale information, these methods tend to rely intensively on a single scale and under-fitting the others, likely attributed to the imbalanced nature and inherent greediness of multi-scale learning. To alleviate the optimization imbalance, we present MUSE, a multi-scale representation learning framework based on a variant expectation maximization to optimize different scales in an alternating procedure over multiple iterations. This strategy efficiently fuses multi-scale information between atomic structure and molecular network scale through mutual supervision and iterative optimization. MUSE outperforms the current state-of-the-art models not only in molecular interaction (protein-protein, drug-protein, and drug-drug) tasks but also in protein interface prediction at the atomic structure scale. More importantly, the multi-scale learning framework shows potential for extension to other scales of computational drug discovery.

Interactions between proteins, drugs, and other biomolecules play a crucial role in various biological processes[1–3]. Understanding these interactions is essential for deciphering the molecular mechanisms underlying biological processes and developing new therapeutic strategies[4]. However, the massive growth in demand and cost associated with experimental interactions calls for computational tools for automated prediction and understanding of interactions between biomolecules[5].

Many computational methods have been developed for studying the interactions between biomolecules[6–9]. Predicting these interactions purely from structures is one of the most important challenges in structural biology[10,11]. The structural-based methods aim to promote our understanding of the patterns of interactions among residues/atoms at the atomic structural scale (intra-molecular scale). Unfortunately, these methods often lead to inferior performance when high-quality molecular structural features are not available[9]. On the other

[1]School of Computer Science and Engineering, Sun Yat-sen University, Guangzhou, China. [2]Global Institute of Future Technology, Shanghai Jiao Tong University, Shanghai, China. [3]Key Laboratory of Machine Intelligence and Advanced Computing (MOE), Sun Yat-sen University, Guangzhou, China. [4]State Key Laboratory of Oncology in South China, Sun Yat-sen University, Guangzhou, China. ✉e-mail: yutong.lu@nscc-gz.cn; shuangjia.zheng@sjtu.edu.cn; yangyd25@mail.sysu.edu.cn

hand, network-based methods analyzed the topology of the molecular networks to infer potential new interactions[12,13]. Similarly, these methods don't perform well due to ignorance of protein structures. Thus, accurate predictions need to capture multi-scale hierarchical and complementary information.

To model the joint distribution over the multi-scale of protein and drug interactions, a few attempts have been developed[14-17]. An intuitive approach for learning multi-scale representations is to combine the molecular graph with an interaction network and optimize them jointly. For example, HIGH-PPI[16] is the first to jointly optimize a hierarchical graph, including the PPI network (outside-of-protein view) and the protein graph (inside-of-protein view) for protein-protein interactions. MIRACLE[15] proposed a contrastive learning strategy to integrate the multi-view information of drug-drug interactions. ScanNet[18] also integrates information from multiple scales (atom, amino acid) to improve the prediction of protein-protein binding sites. However, due to the imbalanced nature and inherent greediness of multi-scale learning[19-21], these models often intensively rely on a single scale, allowing it to learn faster. This imbalanced nature prevents these approaches from effectively leveraging all informative scale-related information and often results in worse generalization. Furthermore, an effective multi-scale framework needs not only capture the rich information within different scales but also faithfully preserve the underlying relation in between.

In this study, we present MUSE, a multi-scale representation learning framework based on a variant expectation maximization[22], which can effectively integrate multi-scale information for learning. In contrast to existing methods that rely heavily on single-scale information, MUSE effectively addresses the optimization imbalance in multi-scale learning through mutual supervision and iterative optimization. Extensive experiments have shown that MUSE is superior not only for predicting molecular interactions (including protein-protein, drug-protein, and drug-drug interactions) but also for predicting molecular binding interfaces. Such a multi-scale framework has potential for applications at more scales, including atomic and amino acid scales, due to its robustness and scalability.

## Results

### MUSE: A balanced expectation-maximization learning framework to learn protein and drug multi-scale information

As shown in Fig. 1, MUSE is a multi-scale learning method that integrates both molecular structure modeling and interaction network learning of protein and drug through a variational expectation-maximization (EM) framework. The EM framework optimizes two modules, the expectation step (E-step) and the maximization step (M-step), in an alternating procedure over multiple iterations[20,23]. During the E-step, MUSE utilizes the structural information of each biomolecule to learn an effective structural representation for training with known interactions and augmented samples in the M-step. It takes the pair of protein and drug with their atom-level structural information as input and augments with the predicted interactions from the M-step. The M-step takes the molecule-level interaction network, the structural embeddings, and predicted interactions from the E-step as the input, and also outputs the predicted interactions. This iterative optimization between the E-step and the M-step ensures the capturing of both the molecular structures and network information interactively, with different learning rates at two scales. The mutual supervision ensures that each scale model learns in an appropriate manner, enabling the utilization of effective information at different scales. This framework will be demonstrated on several multi-scale tasks for interactions between proteins and drugs. We additionally analyzed that MUSE mitigates the imbalanced characteristics in multi-scale learning and effectively integrates the hierarchical and complementary information from different scales.

### Leveraging atomic structure information for improving predictions at the molecular network scale

To evaluate our method, we first utilized MUSE to integrate atomic structural information to improve molecular network scale predictions. As shown in Fig. 2a, MUSE achieved state-of-the-art performance consistently on the three multi-scale interaction predictions tasks, including protein-protein interactions (PPI)[16,24], drug-protein interactions (DPI)[11,25], and drug-drug interactions (DDI)[15,26].

Specifically, on the PPI dataset, MUSE outperformed all existing models including single-scale (DrugVQA[7] and TAG-PPI[27]), and multi-view methods (GNN-PPI[24] and HIGH-PPI[16]). TAG-PPI and DrugVQA, which solely focus on the atomic structure scale, achieved the poorest performance as they neglected the molecular network scale information. Our model showed substantial improvements over the strongest baseline HIGH-PPI, which also incorporates multi-scale information for enhanced predictions, with an increase of 13.81% in the BFS split, 13.06% in the DFS split, 7.69% in the Random split, (Fig. 2c and Fig. S1). Furthermore, our model also showed improvements over the ablation study MUSE-Joint, which integrates two scale models and optimizes them jointly with multiple iterations, attributed to the efficient utilization of the EM framework. The MUSE-Joint is also slightly better than HIGH-PPI, as HIGH-PPI does not optimize the structural information jointly for predictions.

Similar results have been shown on the DPI and DDI datasets. MUSE achieved the AUPRC of 0.998 and 0.922 and the AUROC of 0.993 and 0.915 respectively on the DDI and DPI datasets (Supplementary Tables S2, S3). When compared with state-of-the-art methods, MUSE showed improvements of 5.05% on the DDI dataset (over CGIB[28]) and 2.67% on the DPI dataset (over ConPlex[11]). Interestingly, the baseline method (MIRACLE[15]), which directly combines information from different scales in GNN, underperforms the recent structure-based baseline method CGIB on most metrics. These results indicated the importance of the multi-scale integration strategy, and a sub-optimal strategy might even lead to worse performance than single-scale models.

### Improving predictions at the atomic structure scale from the molecular network scale

In addition to leveraging atomic structural information to improve molecular network scale prediction, we further investigated the ability of MUSE on the learning and prediction of structural properties at the atomic structural scale, including the prediction of interface contacts[29-31] and binding sites[18,32,33] related to protein-protein interactions (PPI).

To evaluate the prediction of protein inter-chain contact, MUSE was compared with the state-of-the-art methods on the DIPS-Plus benchmark[34]. As shown in Fig. 3a, MUSE consistently outperformed all other methods[31,35-37], validating its effectiveness and adaptability in the atomic structural predictions. To be specific, our model achieved the AUROC of 0.92 and the highest precision of 0.26, 0.25, and 0.23 for P@10, P@L/10, and P@L/5, respectively, significantly outperforming the best method, DeepInteract, with AUROC of 0.90 and precision values of 0.20, 0.19, and 0.17. This improvement could likely be attributed to the EM training paradigm in MUSE, which is capable of learning geometric representations (E-step) and interaction patterns (M-step) of proteins in an iterative optimization process. We also highlighted the capabilities of MUSE by examining three examples from the testing set. As shown in Fig. 3b (and Supplementary Fig. S2), MUSE accurately predicted contact interfaces with an average precision of 0.711, which is significantly better than DeepInteract's 0.592 and GLINTER's 0.019. Furthermore, the predicted contact map of MUSE leads to better docking results upon integration of the Kabsch algorithm (RMSD: 4.33). In contrast, alternative methods struggle to generate accurate binding structures, resulting in a relatively high RMSD (DeepInteract: 8.15 and GLINTER: 19.90).

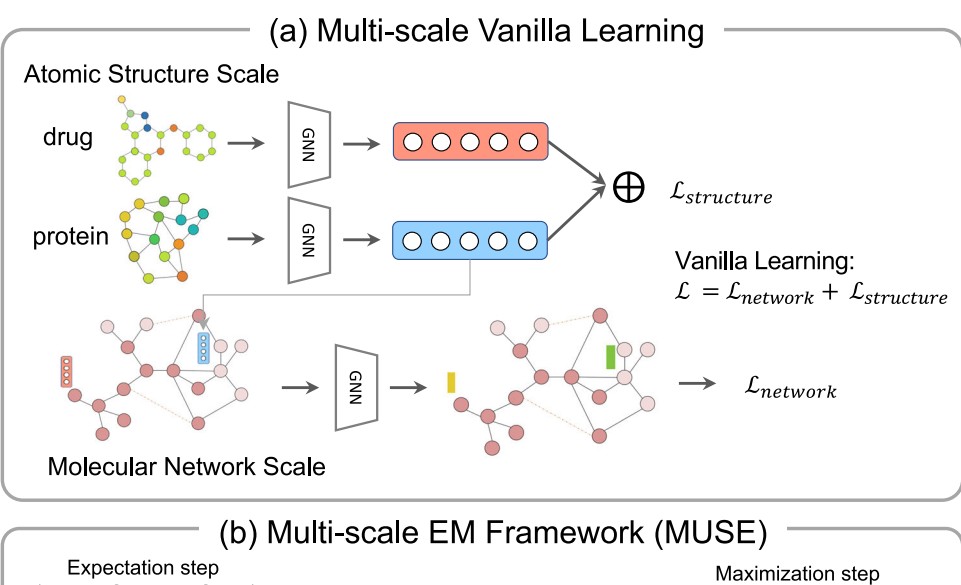

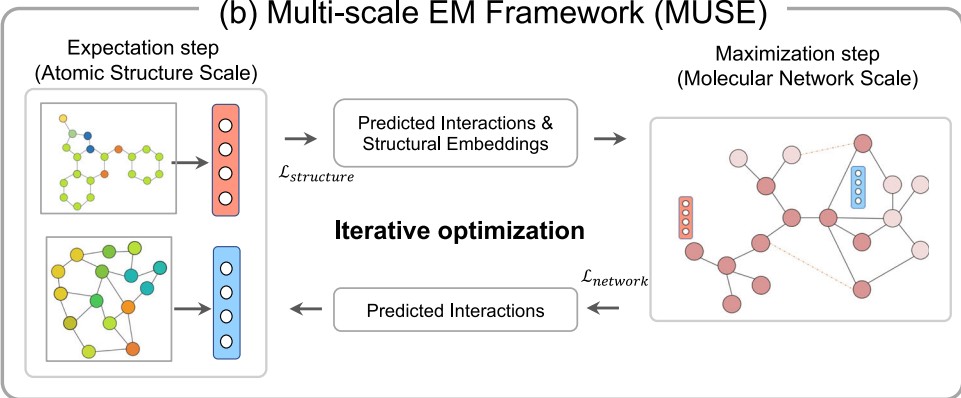

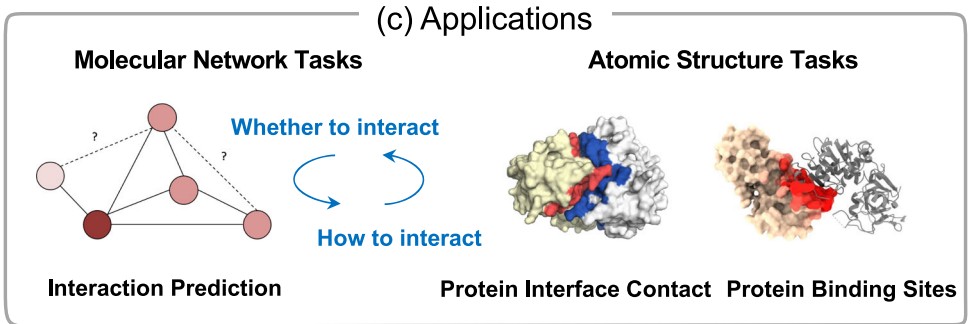

**Fig. 1 | Illustration of the MUSE framework and its applications. a** The vanilla approach for learning multi-scale representations is to combine the two single-scale models and optimize them jointly. **b** The EM framework (MUSE) for Multi-scale Learning. The Expectation step trains a model with structural information of protein and drug as input to fit the known and pseudo interactions (the predicted interactions from the M-step except the first iteration). The updated interactions and structural embeddings were input to the M-step, where the molecular network was constructed to maximize the prediction of the known interactions and the predicted interactions from the E-step. The updated interactions were sent to E-step for new iterations. **c** MUSE is generalizable and applicable to multiple prediction tasks: predicting different types of molecular interactions and molecular binding interfaces.

MUSE was further evaluated to predict whether the residues are directly involved in protein-protein interactions[18,33]. As shown in Fig. 3c and Supplementary Table S3, on the ScanNet[18] benchmark, MUSE achieved a Median AUPRC of 0.811 and a Median AUROC of 0.938 for the full test set, 1.76% and 0.969% better than the second best method PeSTo[33]. The improvements demonstrate that the learning of molecular network scale in MUSE could provide valuable insights into the predictions at the atomic structural scale. MUSE also performed best on another benchmark dataset developed by Masif-site[10], 2.23% and 1.40% better than PeSTo and MUSE-Joint, respectively. These improvements are in line with our expectation as the effective integration of multiple scales (atom, amino acid, molecule) could efficiently detect protein-protein binding sites. Visualizations of MUSE predictions for representative examples (Supplementary Fig. S3) illustrate that the binding sites are correctly identified (0.978 and 0.977 accuracy).

### Mitigating the imbalanced characteristics of multi-scale learning through iterative optimization

To investigate why MUSE achieved the superior performance of multi-scale representations, we analyzed the learning ability of MUSE for the imbalanced characteristics of multi-scale learning. We hypothesized that the slow-learning scale's updating direction is severely disturbed by the dominant one, making it hard to exploit its features for accurate predictions[19,38].

To indicate the information utilization at different scales, we defined the information utilization rate as the proportion of accuracy

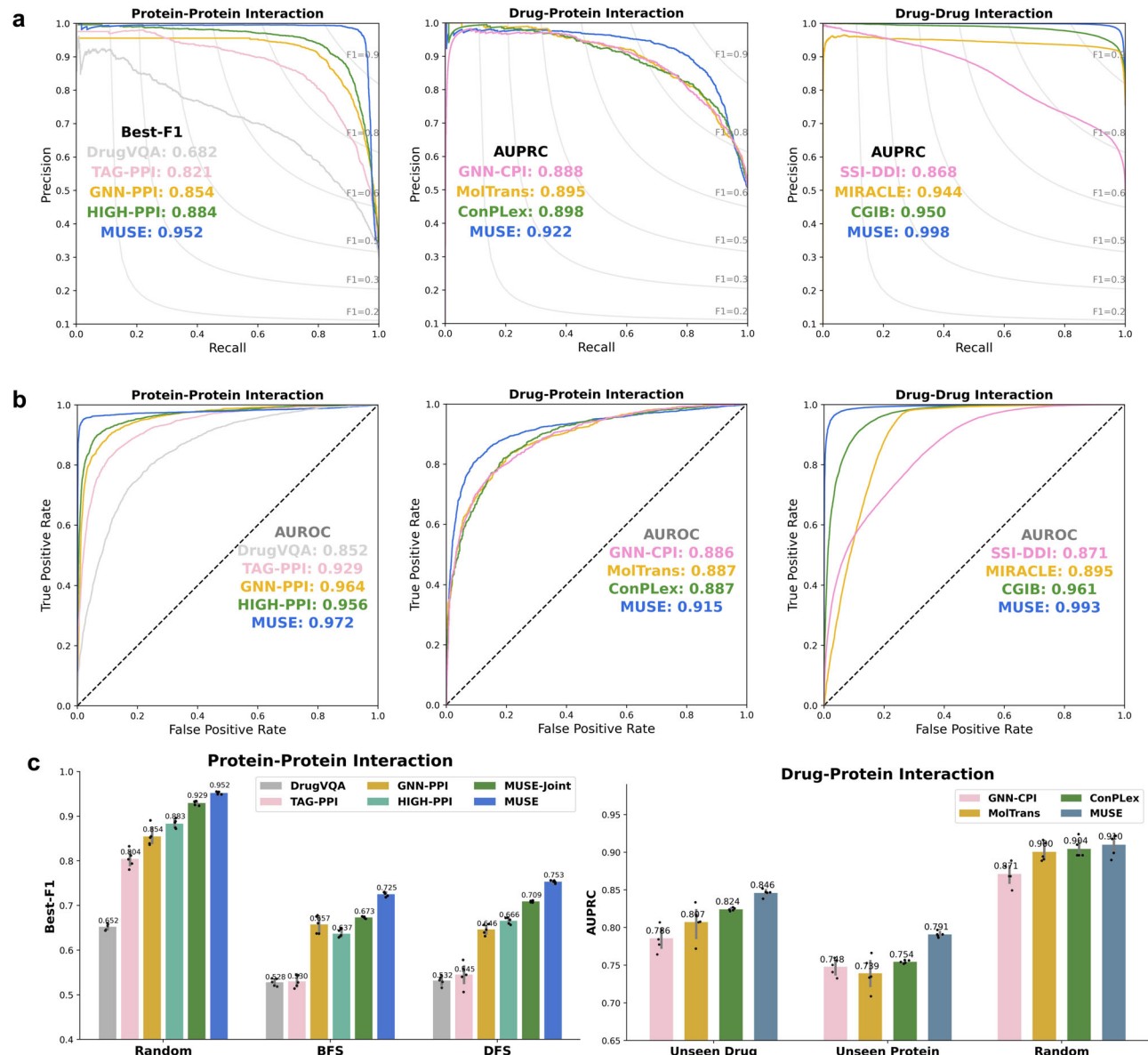

**Fig. 2 | Performance of MUSE in predicting protein and drug interactions.**
**a** Precision-recall curves of PPI prediction on SHS27k, DPI prediction on BioSNAP, and DDI prediction on DeepDDI, showing the performance of MUSE compared to state-of-the-art baselines. **b** Receiver Operator characteristic curves of PPI prediction on SHS27k, DPI prediction on BioSNAP, and DDI prediction on DeepDDI, showing the performance of MUSE compared to state-of-the-art baselines.

**c** Barplot shows the best micro-F1 scores (Best-F1) or AUPRC of baseline, ablation study MUSE-Joint and MUSE predictions respectively on PPI predictions (Random, BFS and DFS splits), DPI predictions (Unseen Protein or Drugs) and DDI predictions. Error bars and the the corresponding data points (as dot plots) represent standard deviation of the mean under 5 independent runs. Source data are provided as a Source Data file.

changes achieved by single-scale models relative to the final multi-scale model, similar to previous studies[19,38]. As shown in Fig. 4a, for HIGH-PPI, the utilization rate is 0.009 for the atomic structure scale and 0.191 for the molecular network scale, leading to a ratio of 21.22. These indicated that HIGH-PPI was dominated by the molecular network scale. Similarly, MIRACLE and other methods were also harmed by molecular network scale information. In contrast, the utilization rates of MUSE for the atomic structure and molecular network scales rose to 0.103 and 0.318, with a ratio of 3.08. These results indicated that MUSE efficiently alleviates the inhibition on the molecular network scale and fully exploits the structural properties in multi-scale learning. The different contributions of different scales to the learning objective are due to the natural imbalance that exists in the current dataset.

To explore the contribution of iterative optimization, we could see that MUSE did not perform as well as other baseline methods in the

beginning but outperformed them after iterations (Fig. 4b). The continuous increase in accuracy (black curve) shows how MUSE continuously alleviates the imbalance characteristics of multi-scale learning and improves its multi-scale utilization. The learned representations by MUSE classify the interaction type between protein and protein while both the molecular network model (Fig. 4c) and MUSE without iterative optimization (MUSE-Joint) have a small number of samples mixed. Specifically, Case (1) demonstrates that MUSE-Joint inhibits the utilization of structural information for predictions, while MUSE alleviates this inhibition and fully utilizes these structural features. On the other hand, Case (2) illustrated that MUSE-Joint is disturbed by the other scale during training, whereas MUSE does not encounter this issue.

In summary, MUSE effectively mitigates the imbalance characteristics and greedy learning in multi-scale learning, ensuring the

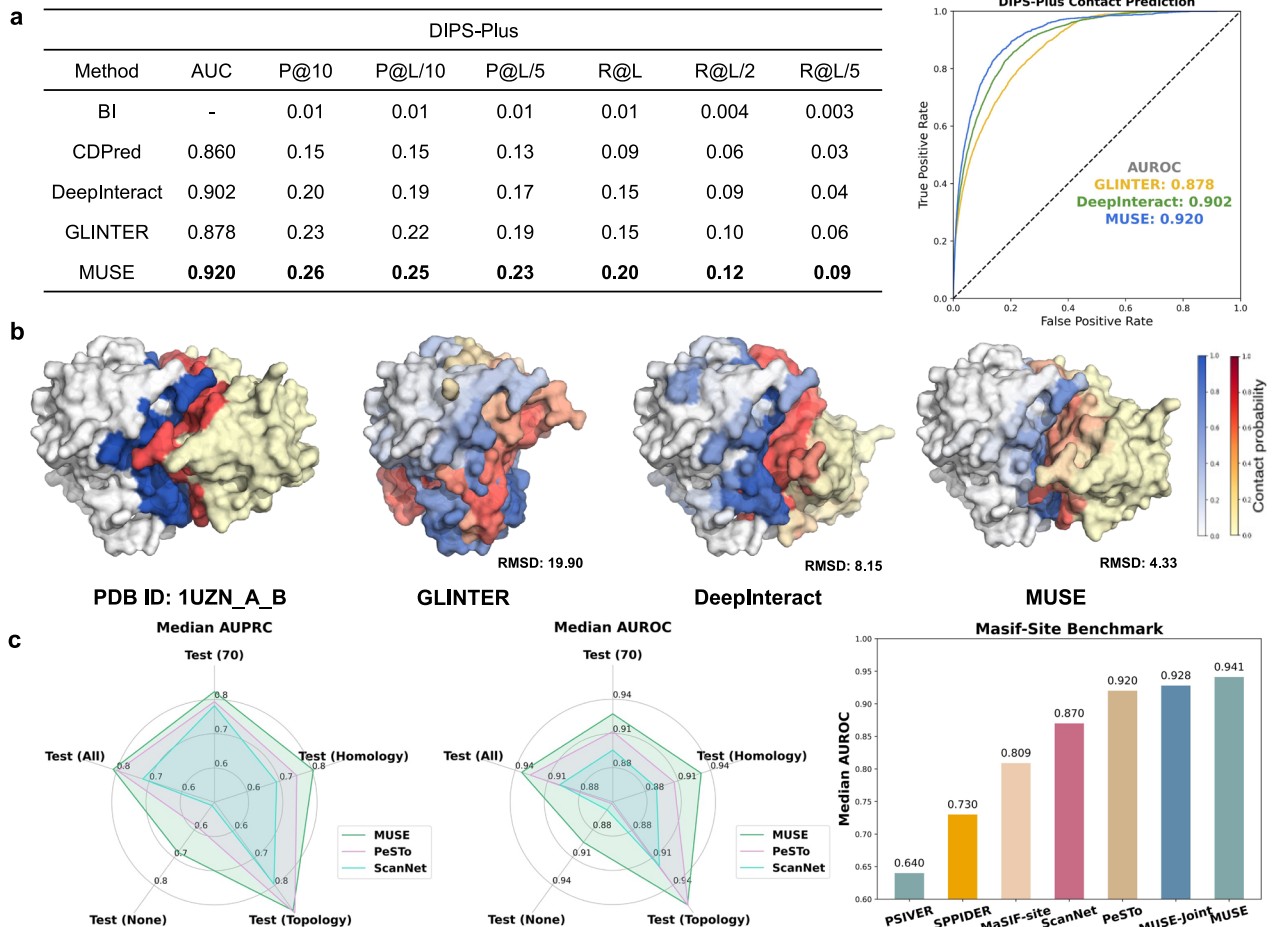

**Fig. 3 | Performance of MUSE at the atomic structure scale. a** The average top-k precision, recall, and AUROC metrics for different protein-protein interaction prediction methods on the DIPS-Plus benchmark. Five predictors are compared: BIPSPI, DeepInteract, GLINTER, CDPred, and MUSE. The best results are marked as **bold**. **b** The ground truth and predictions for PDB entry 1UZN are presented. For each method, we display the surface of the predicted and ground truth ligand relative to the ground truth receptor. We employed the Kabsch algorithm to dock the two proteins based on the contact map predicted by each method and calculated the Root Mean Square Deviation (RMSD) for the ground truth. **c** MUSE achieves state-of-the-art performance on a broad range of test sets in the PPBS dataset compared with state-of-the-art baselines and an ablation study MUSE-Joint. Source data are provided as a Source Data file.

comprehensive utilization of information at different scales during training. Furthermore, the experiment of utilization rate analysis enables us to have a concrete look at what the model has learned and demonstrate that using MUSE to balance the models' learning from different scales enhances generalization.

### Visualization and interpretation of the learned multi-scale representations

To better understand the learned multi-scale representations, we investigated the learned multi-scale representation by MUSE from different perspectives, including (1) the capability of MUSE to capture the atomic structure information (i.e. structural motifs and embeddings) involved in PPI, and (2) the mutual supervision between the learned atomic structure and molecular network representations.

As an example of binding site prediction (PDB id: 3CQQ-A), MUSE can accurately identify the residues belonging to the binding sites (Fig. 5a) (97.7% accuracy). This demonstrated that the mutual supervision in MUSE helps the atomic structure scale model to learn key substructures related to interactions. The learned atomic structural representations (Fig. 5b) confirmed that our learned representation aids in discriminating the interaction types while HIGH-PPI showed a distribution close to the random. This agreed that HIGH-PPI didn't well utilize the structural information with a low atomic structure information utilization rate (0.009).

To illustrate the role of mutual supervision, we conducted an ablation study to investigate the effect of the pseudo-labels predicted by the atomic structure scale on the molecular network scale. We computed the best-F1 scores at different thresholds for pseudo-labeling on the PPI dataset (box plots, Fig. 5c). Without implementing mutual supervision ($t = 0$), the model achieved a best-F1 of 0.908, only marginally outperforming the baseline method (0.886, HIGH-PPI). As the threshold value of $t$ increases, the Best-F1 score of MUSE improves rapidly. This improvement is attributed to the addition of more pseudo interactions to the PPI network, mitigating the incompleteness of the network[39]. Subsequently, the performance gradually decreases as the threshold $t$ increases, indicating that the predicted pseudo interactions become increasingly noisy when $t > 0.4$.

## Discussion

MUSE unifies two scales of biomolecules, atomic structure scale and molecular network scale, into a multi-scale framework. The iterative optimization and mutual supervision process partially overcome the greedy nature of multi-scale learning, and thus significantly improve the multi-scale representation learning. The method was shown effective not only for predicting molecular interactions but also for predicting molecular binding interfaces. These advantages distinctly empower MUSE for the multi-scale learning of proteins and drugs that could be extended to other multi-scale tasks.

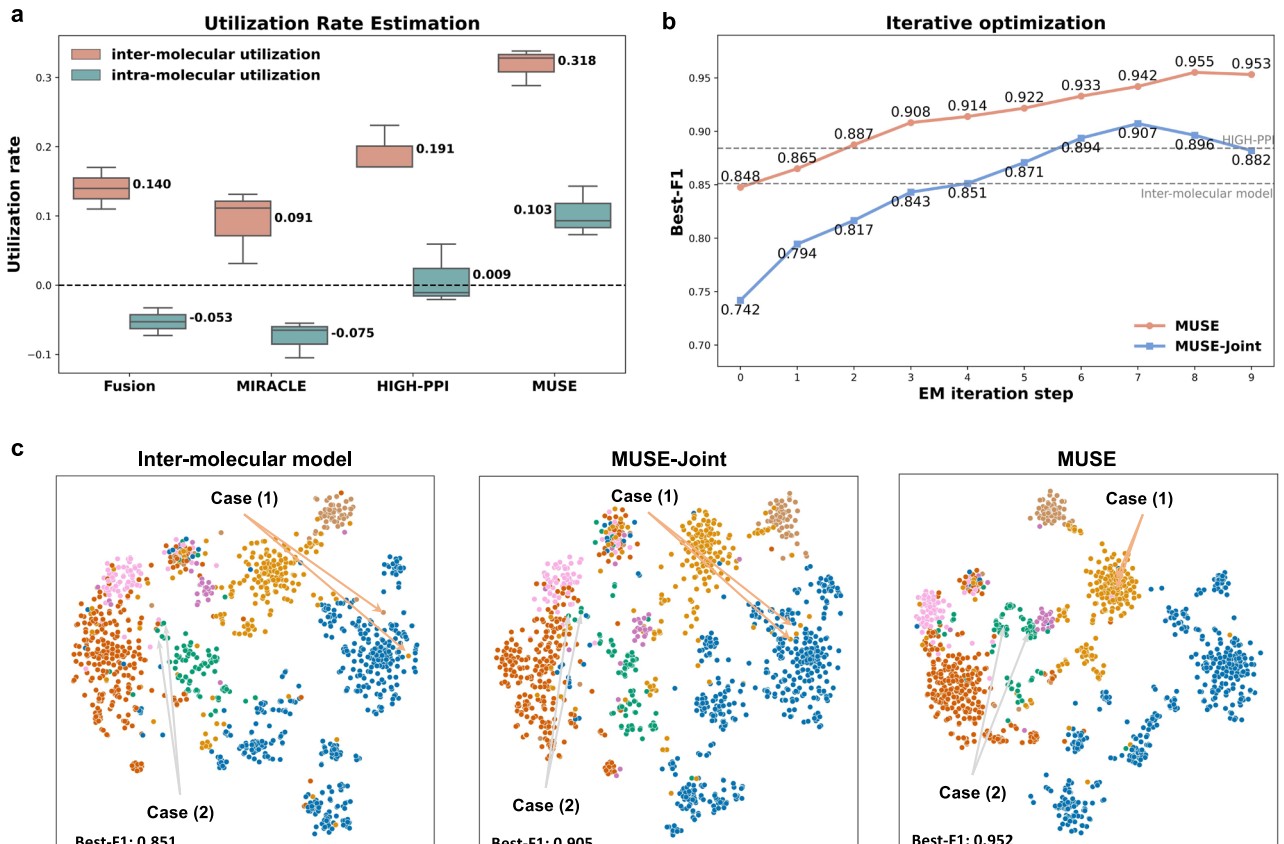

**Fig. 4 | Analyzing the imbalanced characteristics in multi-scale learning. a** The boxplot (5 runs with independent seeds) of utilization rate in each scale for different multi-scale models, demonstrating that MUSE efficiently alleviates the inhibition on molecular network scale and fully exploits the atomic structural properties in multi-scale learning. For boxplots, the center line represents the median, upper and lower edges represent the interquartile range, and the whiskers represent 0.5 × interquartile range. **b** The convergence curves of MUSE and MUSE-Joint on the PPI dataset during the EM optimization iteration steps. The black curve

in iterative optimization shows how MUSE continuously alleviates the imbalance characteristics of multi-scale learning and improves its multi-scale utilization. **c** Visualization and Case Study on the interaction predictions of molecular network model, MUSE-Joint and MUSE. Case (1) represents the examples in which MUSE-Joint cannot utilize other scale information for prediction, and Case (2) represents the examples in which MUSE-Joint is disturbed by other scale learning. MUSE classifies these interaction types between protein and protein clearly. Source data are provided as a Source Data file.

With the continuously growing of multi-scale data, the integration of data across different scales is becoming increasingly critical. Unfortunately, achieving this integration is challenging, as the straightforward approaches are often dominated by individual scale due to the imbalanced nature for different scales and the inherent greediness of deep neural networks. Our approach provides an effective perspective for integrating unbalanced multi-scale data. We show the efficacy of MUSE across various interaction prediction tasks, highlighting the superiority of integrating atomic structure scale information into molecular interaction predictions. Furthermore, the molecular network scale learning within MUSE offers valuable insights for further optimizing the atomic structure scale model to enhance protein representations, as indicated by the improvements in the atomic structure scale tasks.

While MUSE has demonstrated state-of-the-art performance in our benchmarks, there remains potential for enhancing its ability to handle noisy and incomplete multi-scale downstream tasks. This might be combined to incorporate prior knowledge through a knowledge graph and explainable AI techniques. On the other hand, our conceptual multi-scale framework also exhibits potential for extension to other scales of computational drug discovery. For example, MUSE could be applied to the atom and amino acid scale in protein representation learning, deepening our understanding of the multiple structural scales of protein. We hope that its broad applicability and scalability to other scales will contribute to the drug discovery of effective therapeutics.

## Methods

### Datasets with multi-scale learning

We have evaluated our framework on three multi-scale interaction prediction benchmarks, the protein inter-chain benchmark (DIPS-Plus) and the protein binding sites benchmark (Scannet). The statistics of these datasets are presented in Supplementary Information.

Firstly, we evaluated our framework on three multi-scale interaction prediction benchmarks, i.e., protein-protein interaction predictions (SHS27K), drug-protein interaction predictions (BioSNAP), and drug-drug interaction predictions (DeepDDI). The SHS27K PPI dataset contains SHS27k (sub-dataset of STRING[40]) with 6660 protein-protein pairs (PPIs) and 1533 human proteins with native protein structures. These PPIs are divided into 7 types, namely reaction, binding, post-translational modifications (ptmod) activation, inhibition, catalysis, and expression, which contain 15,056 positive interaction types and 31,564 negative interaction types. The native protein structures are obtained from PDB (https://www.rcsb.org/), in line with previous works[16]. The BioSNAP dataset, obtained from[25], consists of only positive drug-target interactions. The negative drug-protein interactions were generated by randomly selecting an equal number of protein-drug pairs[15,28]. We ensure that our data splits and sampling methods align with those used in previous work[11]. Herein, we used the molecular 2D graph for drugs and the protein structures are obtained from the pre-trained model ESMFold[41]. For the DeepDDI dataset, we used 192,284 pair-wise drug-

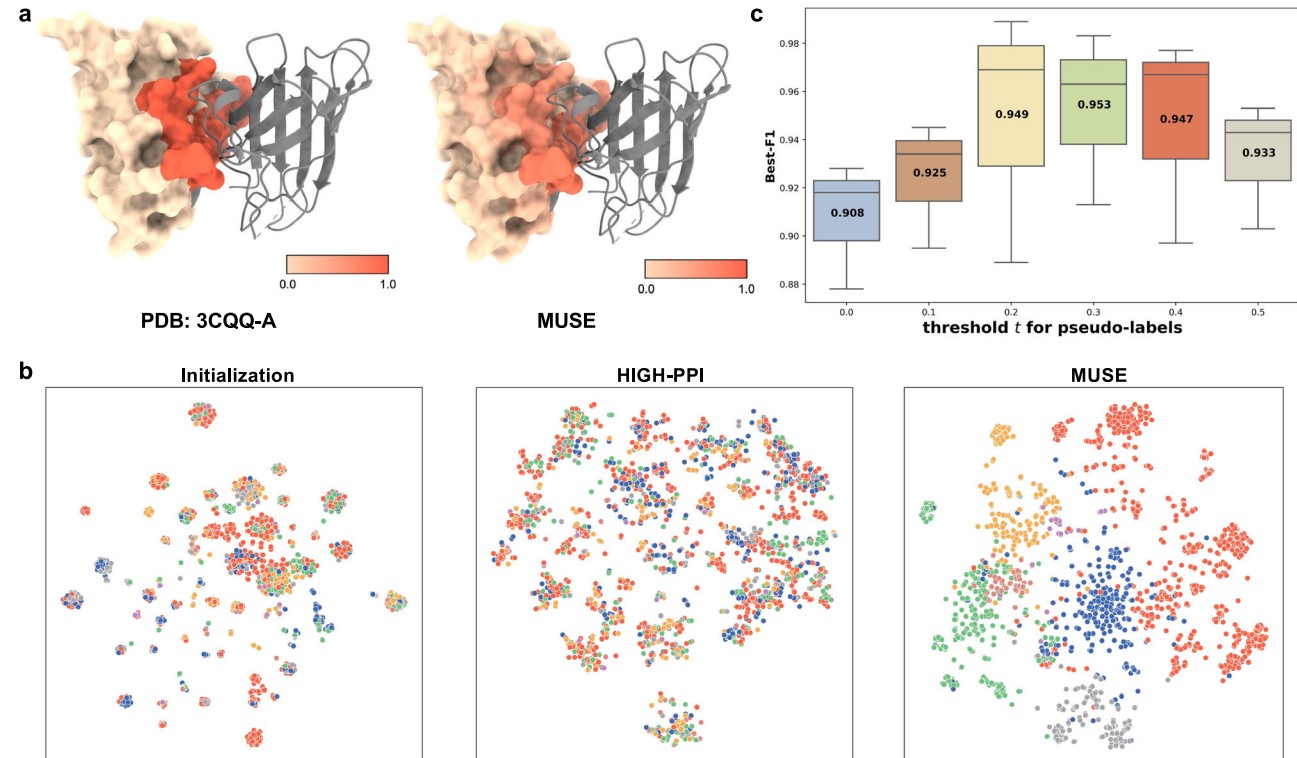

**Fig. 5 | Visualization and interpretation of the multi-scale representations.**
**a** Left: Depiction of a complex protein (protein, PDB id: 3CQQ-A). Right: Residue importance of the query protein learned from MUSE with coloring ranging from low (yellow) to high (red). **b** Two-dimensional projection of the learned atomic structure scale representation using t-SNE. Each point corresponds to an interaction. Coloring is based on the interaction types. The learned atomic structural representation of MUSE is effective for label prediction while the representation learned by HIGH-PPI is quite close to the random initialization. **c** The molecular network scale performance (Best-F1) of MUSE with the different threshold of pseudo labels (3 runs with independent seeds) from the atomic structure model indicates that the mutual supervision in MUSE leads to superior performance. For boxplots, the center line represents the median, upper and lower edges represent the interquartile range, and the whiskers represent 0.5 × interquartile range. Source data are provided as a Source Data file.

drug interactions and their polypharmacy side-effect information extracted from DrugBank[42].

When predicting the protein interface contacts at the atomic structure scale, we chose to use DIPS-Plus[34], a large protein complex structures dataset mined from the Protein Data Bank and tailored for rigid body docking. Following the previous study[31], we select 32 homodimers and heterodimers to evaluate the performance of our model in predicting interface contacts. After removing proteins with ≥ 30% sequence identity with the test datasets, 15,618 and 3548 binary complexes are left for training and validation.

For the prediction of protein-protein binding sites (PPBS), we obtained 41,466 distinct PDB files, involved in 240,506 protein-protein interfaces from the Dockground database[43]. Following Tubiana et al.[18], we investigated the impact of homology between train and test set examples on generalization of our framework and baseline models and therefore grouped validation and test examples into four subgroups based on their degrees of homology: (1) Val/Test 70% (at least 70% sequence identity with at least one train set example), (2) Val/Test homology (at most 70% sequence identity with any train set example), (3) Val/Test topology (at least one train set example with similar protein topology), and (4) Val/Test none (none of the above). The additional independent testing set is composed of clusters containing any of the 53 subunits from the MaSIF-site benchmark dataset.

### Notations and Problem Formulation

Suppose that we have a multi-scale network $\mathcal{N}$ which is presented by $\mathcal{N} = \{\mathcal{G}, \mathcal{L}\}$ where $\mathcal{G} : = \{G_i\}_{i=1}^{N}$ is the set of the biomolecular graph $G$ and $\mathcal{L} : = \{L_{i,j}\}_{(i,j)}^{M}$ is the set of the known interaction link $L$ between

biomolecules. For each node $G$ in $\mathcal{N}$, we denote the biomolecular structural graph $G = \{\mathcal{V}, \mathcal{E}\}$, where $\mathcal{V}$ is the set of atoms $v \in \mathcal{V}$ and $\mathcal{E}$ is the set of edges/bonds $e \in \mathcal{E}$, so the known interaction links between biomolecules also can be denoted as the pairs of molecular atomic structural graphs: $\mathcal{D} : = \{(G_i, G_j)\}_{(i,j)}^{M}$. Herein, we study the problem of link prediction on $\mathcal{N}$, for predicting the labels $Y_{\mathcal{L}_U}$ of the unobserved (test-set) interactions $\mathcal{L}_U$ with a few observed labels $Y_{\mathcal{L}_V}, \mathcal{L} = \mathcal{L}_V \cup \mathcal{L}_U$. We also defined the adjacency matrix in the network $\mathcal{N}$ as $A \in \mathbb{R}^{(N \times N)}$ where $A_{i,j} = 1$ if $(i, j) \in \mathcal{L}_V$ and $A_{i,j} = 0$ otherwise.

### Variational Expectation-Maximization (EM) Framework

At the core of our method is combining the atomic structure scale and molecular network scale models for link prediction learning on multi-scale networks with a variational EM framework. Given the observed variables $Y_{\mathcal{L}_V}$, unobserved variables $Y_{\mathcal{L}_U}$, and model parameters $\theta, \phi$, our framework tries to maximize the log-likelihood function of the observed interaction labels, i.e. $\log p_\theta(Y_{\mathcal{L}_V}|\mathcal{G}, A)$. It is computationally intractable to compute this log-likelihood as it requires integration over all combinations of object labels, i.e. $\log \prod_{l \in \mathcal{L}_V} p_\theta(Y_l|\mathcal{G}, A)$. Thus we instead optimize the evidence lower bound (ELBO) of the log-likelihood function:

$$\mathcal{O}(\theta) = \log p_\theta(Y_{\mathcal{L}_V}|\mathcal{G}, A) \geq \mathbb{E}_{q_\phi(Y_{\mathcal{L}_U}|\mathcal{G}, A)}\left[\log p_\theta(Y_{\mathcal{L}_V}, Y_{\mathcal{L}_U}|\mathcal{G}, A) - \log q_\phi(Y_{\mathcal{L}_U}|\mathcal{G}, \mathcal{D})\right]$$
(1)

where $q_\phi(Y_{\mathcal{L}_U}|\mathcal{G}, \mathcal{D})$ can by any variation distributions over $Y_{\mathcal{L}_U}$, and the equation holds when $q_\phi(Y_{\mathcal{L}_U}|\mathcal{G}, \mathcal{D}) = p_\theta(Y_{\mathcal{L}_U}|Y_{\mathcal{L}_V}, \mathcal{G}, A)$. The ELBO is also challenging to directly derive the maximum likelihood estimator via the EM algorithm. Therefore, we use a variational approximation of the EM algorithm to optimize the lower bound by iteratively alternating

between optimizing the distribution $q$ (i.e., E-step) and the distribution $p$ (i.e., M-step).

In the variational E-step, the goal is to fix $p_\theta$ and update $q_\phi$ to minimize the KL divergence between $q_\phi(Y_{\mathcal{L}_U}|\mathcal{G},\mathcal{D})$ and $p_\theta(Y_{\mathcal{L}_U}|Y_{\mathcal{L}_V},\mathcal{G},A)$. In the M-step, we aim to update $p_\theta$ to maximize the below likelihood function:

$$\mathcal{O}(\theta) = \mathbb{E}_{q_\phi(Y_{\mathcal{L}_U}|\mathcal{G},\mathcal{D})}[p_\theta(Y_{\mathcal{L}_U}, Y_{\mathcal{L}_V}|\mathcal{G},A)] \qquad (2)$$

To prevent a computational complexity of $p_\theta$ in the EM algorithm, we utilized the following pseudo-likelihood function:

$$\mathcal{O}_{PL}(\theta) \triangleq \mathbb{E}_{q_\phi(Y_{\mathcal{L}_U}|\mathcal{G},\mathcal{D})}\left[\sum_{L_{ij}\in\mathcal{L}_V} \log p_\theta(Y_{L_{ij}}|\mathcal{G},A,Y_{\mathcal{L}\setminus L_{ij}})\right]$$
$$\triangleq \mathbb{E}_{q_\phi(Y_{\mathcal{L}_U}|\mathcal{G},\mathcal{D})}\left[\sum_{L_{ij}\in\mathcal{L}_V} \log p_\theta(Y_{L_{ij}}|\mathcal{G},A,Y_{NB(L_{ij})})\right], \qquad (3)$$

where $NB(L_{ij})$ is the neighborhood information around interactions $L_{ij}$.

Next, we introduce how we apply the framework to link prediction in the multi-scale network by instantiating the $p$ and $q$ distributions with atomic structure scale $q_\phi$ and molecular network scale $p_\theta$ models respectively.

### E-Step: Atomic Structure scale Modeling

The variational E-step aims to update the variational distribution $q_\phi(Y_{\mathcal{L}_U}|\mathcal{G},\mathcal{D})$ to approximate the true posterior distribution $p_\theta(Y_{\mathcal{L}_U}|Y_{\mathcal{L}_V},\mathcal{G},A)$.

Therefore, we could minimize the KL divergence between the posterior distribution and the variational distribution:

$$KL(q_\phi(Y_{\mathcal{L}_U}|\mathcal{G},\mathcal{D}) \| p_\theta(Y_{\mathcal{L}_U}|Y_{\mathcal{L}_V},\mathcal{G},A)). \qquad (4)$$

We followed the idea of [20,23] to utilize the wake-sleep algorithm[44] for minimizing the reverse KL divergence (refer to Supplementary information A.2 for detailed proofs):

$$\mathcal{O}(q_\phi(Y_{\mathcal{L}_U})|\mathcal{G},\mathcal{D}) = -KL(q_\phi(Y_{\mathcal{L}_U}|\mathcal{G},\mathcal{D}) \| p_\theta(Y_{\mathcal{L}_U}|Y_{\mathcal{L}_V},\mathcal{G},A))$$
$$= \sum_{Y_{\mathcal{L}_U}} q_\phi(Y_{\mathcal{L}_U}|\mathcal{G},\mathcal{D})\left[\log p_\theta(Y_{\mathcal{L}_U}|Y_{\mathcal{L}_V},\mathcal{G},A) - \log q_\phi(Y_{\mathcal{L}_U}|\mathcal{G},\mathcal{D})\right]$$
$$= -KL\left(q_\phi(Y_{\mathcal{L}'}|\mathcal{G},\mathcal{D})\|\frac{\mathcal{F}(Y_{\mathcal{L}'})}{Z}\right) + \text{const} \qquad (5)$$

where $Z$ is a normalization term and $\mathcal{F}(Y_{\mathcal{L}'})$ is the distribution on $Y_{\mathcal{L}'}$:

$$\mathcal{F}(Y_{\mathcal{L}'}) = \mathbb{E}_{q_\phi(Y_{\mathcal{L}_U\setminus L_{ij}}|\mathcal{G},\mathcal{D})}\left[\log p_\theta(Y_{\mathcal{L}_U}|Y_{\mathcal{L}_V},\mathcal{G},A)\right] \qquad (6)$$

Therefore, we no longer need to consider the entropy of $p_\theta(Y_{\mathcal{L}_U}|Y_{\mathcal{L}_V},\mathcal{G},A)$. The KL divergence could be formulated as:

$$KL(q_\phi(Y_{\mathcal{L}_U}|\mathcal{G},\mathcal{D}) \| p_\theta(Y_{\mathcal{L}_U}|Y_{\mathcal{L}_V},\mathcal{G},A)) =$$
$$-\sum_{L_{ij}\in\mathcal{L}_U} \mathbb{E}_{q_\phi(Y_{L_{ij}}|\mathcal{G},\mathcal{D})}\left[\log p_\theta(Y_{L_{ij}}|Y_{\mathcal{L}_V},\mathcal{G},A)\right] + \text{const}. \qquad (7)$$

Herein, to model the distribution of each interaction, we parameterize $q_\phi(Y_{L_{ij}}|\mathcal{G},\mathcal{D})$ with a molecular network scale GNN$_\phi$ for predicting the interaction labels:

$$q_\phi(Y_{L_{ij}}|G,\mathcal{E},Y_{NB(L_{ij})}) = q(Y_{L_{ij}}|\sigma(W_\phi \cdot F(h_i, h_j))), \qquad (8)$$

where the probability of each interaction class is calculated by a softmax/sigmoid classifier $\sigma$ based on the interaction representation $F(h_i, h_j)$. $F$ is the concatenation function to concatenate the node

representations of the linking nodes $i, j$ (i.e., $F = MLP$; $h_{ij} = MLP(h_i \odot h_j)$), and the node representation $h_i$ is learned by an atomic structure scale GNN model, which is denoted as GNN$_\phi$.

To model the GNN$_\phi$, we use the structural graph $G$ of the biomolecules, where we represent $h_i^{(0)} = X_i \in \mathbb{R}^{n\times d_n}$ for the atom attributes with $d_n$ as the feature dimension of atoms and $E_{ij} \in \mathbb{R}^{m\times d_e}$ for the edge/bond attributes with $d_e$ as the feature dimension of edge. Therefore, the molecular structural representation can be denoted as:

$$h_i^{(l)} = f(h_i^{(l)}, f_{AGG}\{(h_i^{(l-1)}, h_j^{(l-1)}, X) : j \in \text{neighbors}(i)\}), \qquad (9)$$

where $f$ and $f_{AGG}$ stand for the message and aggregation functions in the $l$-th layer respectively, $X$ is the transformed node embeddings from the atomic structural graph, and neighbors($i$) denote the neighbor nodes of $i$.

Now, the sole difficulty lies in computing the distribution $p_\theta(Y_{L_{ij}\in\mathcal{L}_U}|Y_{\mathcal{L}_V},\mathcal{G},A))$, which aims to predict the label distribution of an interaction $L_{ij} \in \mathcal{L}_U$ based on the surrounding node features and edge information. However, the labels of the unobserved interactions are not specified. Therefore, we propose to annotate the unobserved interactions with the pseudo-labels predicted by the molecular network scale model GNN$_\theta$, so that we can approximate the distribution as follows:

$$p_\theta(Y_{L_{ij}}|Y_{\mathcal{L}_V},\mathcal{G},A)) = p_\theta(Y_{L_{ij}}|Y_{\mathcal{L}_V},\mathcal{G},A,\hat{Y}_{\mathcal{L}_U\setminus L_{ij}}), \qquad (10)$$

where $\hat{Y}_{\mathcal{L}_U\setminus L_{ij}}$ denote the predicted pseudo interaction labels.

Combining them with the above objective function, we could obtain the final objective function for training the GNN model of atomic structure scale:

$$\mathcal{O}_{q_\phi} = \alpha \sum_{L_{ij}\in\mathcal{L}_U} \mathbb{E}_{p_\theta(Y_{L_{ij}}|Y_{\mathcal{L}_V},\mathcal{G},A,\hat{Y}_{\mathcal{L}_U\setminus L_{ij}})}\left[\log q(Y_{L_{ij}}|\mathcal{G},\mathcal{D})\right]$$
$$+ (1-\alpha) \sum_{L_{ij}\in\mathcal{L}_V} \log q(Y_{L_{ij}}|\mathcal{G},\mathcal{D}), \qquad (11)$$

where $\alpha$ is a hyperparameter. Intuitively, the first term could be viewed as a knowledge-distilling process that teaches the model of atomic structure scale GNN$_\phi$ by forcing it to predict the label distribution based on the predicted pseudo interaction from the molecular network model GNN$_\theta$. The second term is a supervised objective which uses the given labeled interactions for training.

### M-step: Molecular Network scale Modeling

During the M-step, we seek to learn the parameter $\theta$ and update $p_\theta$ to maximize the objective function Eq. (3), which aims to optimize the molecular network model.

Here, we parameterize the conditional distribution $p_\theta$ with another graph neural network (GNN) model on molecular network $\mathcal{N}$ because of its effectiveness:

$$p_\theta(Y_{L_{ij}\in\mathcal{L}_U}|Y_{\mathcal{L}_V},\mathcal{G},A) = p(Y_{L_{ij}\in\mathcal{L}_U}|\sigma(W_\phi \cdot g_{ij})), \qquad (12)$$

where $p_\theta$ is formulated as a categorical distribution, and the probability of each interaction class is calculated by a softmax/sigmoid classifier $\sigma$ based on the interaction representation $g_{ij}$. The interaction representation $g_{ij}$ is derived from the hidden representation in the node level (i.e., nodes $g_i, g_j$) with a concatenation function $F$. The node representation $g_i, g_j$ is learned by a molecular network scale GNN model (GNN$_\phi$), which is used to learn the molecular network information with a message-passing mechanism:

$$g_i^{(l)} = f(g_i^{(l)}, f_{AGG}\{(g_i^{(l-1)}, g_j^{(l-1)}) : j \in \text{neighbors}(i)\}), \qquad (13)$$

where $f$ and $f_{AGG}$ stand for the message and aggregation functions in the $l$-th layer respectively, and neighbors($i$) denote the neighbor nodes of $i$.

To be more specific, we use the $GNN_\phi$ to generate structural representations $g^{(0)}$ as node initial features and feed them into the molecular network model for message passing.

We notice that the objective function Eq. (3) also relies on the expectation with respect to $p_\theta$, which can be approximated by drawing a sample $\hat{Y}_{\mathcal{L}_U}$ from $p_\theta(Y_{\mathcal{L}_U}|Y_{\mathcal{L}_V}, \mathcal{G}, A)$. In other words, we need the atomic structure scale model $GNN_\phi$ to predict a pseudo-label $\hat{Y}_{L_{ij}}$ for each unobserved interaction $L_{ij} \in \mathcal{L}_U$ and combine all the labels $\{\hat{Y}_L\}_{L \in \mathcal{L}_U}$ into $\hat{Y}_{\mathcal{L}_U}$. Therefore, the pseudo-likelihood can be rewritten as follows:

$$
\begin{aligned}
\mathcal{O}_\theta = \beta \sum_{L_{ij} \in \mathcal{L}_U} \log p_\theta(\hat{Y}_{L_{ij}}|\mathcal{G}, A, Y_{\mathcal{L}_V}, \hat{Y}_{\mathcal{L}_U \backslash L_{ij}}) \\
+ (1 - \beta) \sum_{L_{ij} \in \mathcal{L}_V} \log p_\theta(Y_{L_{ij}}|\mathcal{G}, A, Y_{\mathcal{L}_V \backslash L_{ij}}, \hat{Y}_{\mathcal{L}_U}),
\end{aligned}
\tag{14}
$$

where $\beta$ is a hyperparameter that is added to balance the weight of the two terms. Again, the first term can be viewed as a knowledge distillation process that injects the knowledge captured by the model of atomic structure scale $GNN_\phi$ into the molecular network scale model $GNN_\theta$ via all the pseudo-labels. The second term is simply a supervised loss, where we use observed interaction labels for model training.

### Iterative optimization

As illustrated in Fig. 1 (and Supplementary Methods), during each iteration, MUSE first performs the E-step, where it constructs structural graphs for each interaction pair, represented as structural graphs $g_1$ and $g_2$. Structural graph encoders ($f_g$ or $f_d$) are then employed to generate representations of the protein/drug graphs and an interaction predictor is utilized to predict the interaction for the given biomolecule pair. This atomic structure scale model pulls the interacted graph pairs together and models label distributions conditioned on structural attributes. After optimization in E-step, the structural representations and interaction graph are fed into a molecular network scale message passing module. In M-step, information is propagated along interactions in the network for learning network topology and neighbor information. Consequently, MUSE iteratively updates the two modules in the E-step and M-step until the model converges. More importantly, the one-step model provides the interacting pseudo-labels for training the other, as part of mutual supervision. The pseudo-labels generated at the molecular network scale are also used as data augmentation for the atomic structure scale model while those from the atomic structure scale can be used for training the molecular network scale model by adding pseudo edges.

According to the greedy learner hypothesis[19], it is the different speeds at which a neural network learns from different scales that lead to a utilization imbalance. The iterative optimization of the atomic structure and molecular network scale in MUSE enables us to mitigate the hurtful imbalance and achieve stronger generalization on multi-scale representations. Mutual supervision ensures that each scale model learns in the appropriate manner, thereby facilitating the utilization of effective information at different scales.

### Graph neural network for learning protein representations

To evaluate the effectiveness of our EM training paradigm in interaction prediction tasks, we first use a general graph neural network (GNN) to learn graph representations of proteins, consistent with competing methods[16].

Recalling our definition in Sec. 4, we denote the protein structural graph $G = \{\mathcal{V}, \mathcal{E}\}$ where $\mathcal{V}$ is the set of atoms $v \in \mathcal{V}$ and $\mathcal{E}$ is the set of edges $e \in \mathcal{E}$. Herein, $\mathcal{V}$ is a set of amino acid residues in a protein and the $\mathcal{E}$ are obtained from the protein contact map with atomic level 3D

coordinates of proteins. Following[16], we choose the optimal cutoff distance of 10 Å for the presence or the absence of contact between a pair of residues, constructing the adjacency matrix $A_p$. For the feature matrix $h_p^{(0)}$, we use the protein features based on amino acid sequence, refer to[24,45,46]. Each embedding vector represents a set of properties for one amino acid residue, including isoelectric point, polarity, acidity and alkalinity, hydrogen bond acceptor, hydrogen bond donor, octanol-water partition coefficient, and topological polar surface area.

Therefore, GNN outputs the residue-level representations in each block:

$$
h_p^{(l)} = \text{BatchNorm}\left(\text{ReLU}(\hat{A} h_p^{(l-1)} W^{(l)})\right),
\tag{15}
$$

where $\hat{A} = \widetilde{D}^{-1/2} A_p \widetilde{D}^{-1/2}$, $\widetilde{D}$ is the diagonal degree matrix, $W^{(l)}$ is a learnable weight matrix for the $l$-th GCN layer, ReLU, BatchNorm denotes the ReLU activation function and batch normalization, respectively.

Furthermore, we also implemented a geometric graph neural network for learning the effective protein structural information within our EM framework. In this setting, we denote each protein chain as a graph $G$ with edges $\mathcal{E}$ between the $k$-nearest neighbors of its nodes $\mathcal{V}$, with nodes corresponding to the chain's amino acid residues represented by their $C\alpha$ atoms.

Inspired by GVP-GNN[47,48], we leveraged message passing over geometric vectors and scalars, in which messages from neighboring nodes and edges are used to update node embeddings at each graph propagation step as:

$$
h_p^{j \to i} = \text{MPNN}\left(\text{concat}\left(h_v^{(i)}, h_e^{(j \to i)}\right)\right),
\tag{16}
$$

$$
h_v^{(i)} = \text{LayerNorm}\left(h_v^{(i)} + \text{Dropout}\left(\sum_{j:e_{j \to i} \in \mathcal{E}} h_p^{(j \to i)}\right)\right).
\tag{17}
$$

Here, $h_v^{(i)}$ and $h_e^{j \to i}$ are the embeddings of the node $i$ and edge $j \to i$, and $h_p^{j \to i}$ represents the message passed from node $j$ to node $i$ of proteins. For the initial geometric features, we calculate distance, direction, and angle features for each residue as node features, and construct geometric edge features between neighboring residues including distance, direction and orientation.

Finally, we perform the readout operation to obtain the entire graph representation of proteins $H_p$.

$$
H_p = \sum_{v \in \mathcal{V}} Readout(h_v^{(l)}).
\tag{18}
$$

### Graph neural network for learning drug representations

A drug structural graph also can be represented as an attributed graph $G = (\mathcal{V}, \mathcal{E})$, where $|\mathcal{V}| = n$ denotes a set of $n$ atoms (nodes) and $|\mathcal{E}| = m$ denotes a set of $m$ bonds (edges)[49]. We represent $X_v \in \mathbb{R}$ for the node attributes and $E_{uv} \in \mathbb{R}$ for the edge attributes. A graph neural network (GNN) $f^d$ learns to embed an attributed graph $G$ into a feature vector $h_d$. We adopt the Graph Isomorphism Network (GIN) from[50], where the node and edge attributes are propagated at each iteration. Formally, the $l$-th iteration of a GNN is:

$$
h_v^{(l)} = g_U^{(l)}\left(h_v^{(l-1)}, g_{AGG}^{(l)}\{(h_v^{(l-1)}, h_u^{(l-1)}, X_{uv}) : u \in \mathcal{N}(v)\}\right),
\tag{19}
$$

where $h_v^{(l)}$ are the representation of node $v$ at the $l$-th layer, $\mathcal{N}(v)$ is the neighbourhood set of node $v$, $h_v^{(0)}$ is initialised with $X_v$ encoding its atom properties. $g_{AGG}^{(l)}$ stands for the aggregation function and $g_U^{(l)}$ stands for the update function.

After $l$ graph convolutions, $h^{(l)}$ have captured their $l$-hop neighborhood information. Finally, a readout function is used to aggregate

all node representations output by the $l$-th GNN layer to obtain the entire molecule's representation $H_d$:

$$H_d = \sum_{v \in \mathcal{V}} Readout(h_v^{(l)}). \tag{20}$$

## Graph neural network for interaction network message passing

For interaction network learning, we use the graph isomorphism networks (GIN[50]), which has the super-expressive power to capture graph structures and to learn molecular network information. Formally, the interaction network $\mathcal{N}$ is presented as $\mathcal{N} = \{\mathcal{G}, \mathcal{L}\}$, where $\mathcal{G} := \{G_i\}_i^N$ is the set of the biomolecular graph $G$ and $\mathcal{L} := \{L_{i,j}\}_{(i,j)}^M$ is the set of the known interaction links $L$ between biomolecules, which also could be represented as the adjacency matrix $A$. The initial node attributes are obtained from the model of atomic structure scale $H_G^{(0)} = (H_p, H_d)$. Therefore, the molecular network scale model updates the representation of biomolecules in the $l$-th GIN block GNN$_\theta$:

$$
\begin{aligned}
H_v^{(l)} &= GNN_\theta\left(v, A, H_G^{(l-1)}\right) \\
&= BatchNorm\left(ReLU\left(H_v^{(l-1)} + \sum_{u \in \mathcal{N}_{(v)}}\left(H_u^{(l-1)}\right)\right)\right),
\end{aligned} \tag{21}
$$

where $H_v^{(l)}$ denotes the representation of biomolecule $v$ after the $l$-th GIN block, $\mathcal{N}_{(v)}$ is a set of biomolecules adjacent to $v$.

After stacking $l$ GIN layers, node representations $H_G$ are then used to predict existence of each link $(i,j)$:

$$\hat{Y}_{L_{ij}} = \sigma(NCN(h_i^{(l)}, h_j^{(l)})), \tag{22}$$

where NCN is a simple but powerful model to capture pairwise features[39], and $h_i^{(l)}$ is the representation of the node $i$ from $H_G$.

When applying the NCN model for pairwise predictions, we choose to keep the first-order neighbor and set the operator only to intersection:

$$NCN(h_i^{(l)}, h_j^{(l)}) = \sum_{C \in N_{(i)} \cap N_{(j)}} GNN_\theta^{(l)}\left(C, A, H_G^{(l)}\right), \tag{23}$$

where GNN$_\theta$ is the powerful GIN block in molecular network scale to output the final node representations of the last message passing layer, $N_{(i)}, N_{(j)}$ denote the sets of neighboring nodes for biomolecules $i$ and $j$, respectively, and $C$ is the set of common neighbor nodes between biomolecule $i$ and $j$.

## Pseudo likelihood learning in the molecular network scale

The limitation in the performance of link prediction tasks is from the incompleteness of the graph[39]. To alleviate this incompleteness, we adopted the pseudo-likelihood learning in our variational expectation-maximization framework, augmenting the molecular network graph $\mathcal{N}$ with pseudo interactions predicted by the atomic structure scale model GNN$_\phi$.

Formally, when predicting the interaction $L_{ij}$, we aim to utilize the edges $(i, u)$ and $(j, u)$ in the complete graph to compute the common neighbor $C_u$. However, since $L_{iu}$ and $L_{ju}$ are unknown, we let the model of atomic structure scale GNN$_\phi$ output the probability of the existence for $L_{iu}$ and $L_{ju}$ as follows:

$$
\begin{cases}
\hat{Y}_{L_{iu}} = \sigma\left(GNN_\phi(i, u)\right), & \text{if } u \in N_{(j)} - N_{(i)} \\
\hat{Y}_{L_{ju}} = \sigma\left(GNN_\phi(j, u)\right), & \text{if } u \in N_{(i)} - N_{(j)}
\end{cases} \tag{24}
$$

where GNN$_\phi$ represents the GNN model of atomic structure scale, $\hat{Y}_{L_{iu}}$ and $\hat{Y}_{L_{ju}}$ denote the probabilities of the existence of interactions $i \to u$

and $j \to u$, respectively. When the probability exceeds the threshold $1 - t$, the predicted interaction is incorporated into the interaction network graph.

Therefore, the interaction network graph $\mathcal{N}$ has been complemented softly, and then the molecular network model GNN$_\theta$ on the more complete graph to give final predictions.

Furthermore, this strategy has been extended to our iterative optimization process. Starting from the original network $\mathcal{N}^{(0)} = \mathcal{N}$, we iteratively complete the graph to get the $\mathcal{N}^{(k)}$ from $\mathcal{N}^{(k-1)}$ with the help of the prediction of other scale model (i.e. atomic structure scale model) until the final iteration $k$.

## Utilization rate estimation

For a multi-scale model $M$, taking two scales $s_0$ (atomic structure scale) and $s_1$ (molecular network scale) as inputs, its utilization rates for $s_0$ and $s_1$ are defined as

$$u(s_0|s) = \frac{f(M) - f(M_1)}{f(M_1)}, \tag{25}$$

and

$$u(s_1|s) = \frac{f(M) - f(M_0)}{f(M_0)} \tag{26}$$

Here, $M_0$ and $M_1$ represent single-scale models obtained from the multi-scale model $M$, while $f$ denotes the predicted evaluation function. The utilization rate is the relative change in accuracy between the two models within each pair. For example, $u(s_0|s)$ measures the incremental impact of the atomic structure scale $s_0$ on improving the accuracy of label predictions. It is constrained within the range of -1 to 1. A higher utilization rate indicates that the multi-scale model can benefit from incorporating this scale.

## Evaluation metrics

For the three multi-scale interaction prediction benchmarks, i.e. SHS27K, BioSNAP, and DeepDDI, we evaluated their performance using micro-F1 (Best-F1), Area Under the Receiver Operating Characteristic curve (AUROC) and Area Under Precision-Recall Curve (AUPRC). In particular, for the multi-label PPI prediction, the different PPI types in the datasets we used are very imbalanced, so micro-F1 may be preferred.

To evaluate the prediction of protein inter-chain contact, we follow the setting established by[31], where a positive label is assigned to each inter-chain residue pair found within 6 Å of each other in the complex's bound (i.e., structurally-conformed) state. We use Average top-k precision (P@k), Recall (R@k), and Area Under the Receiver Operating Characteristic curve (AUROC) to assess our method in predicting protein-protein interactions and the quality of the generated docking results.

Following the previous studies[18], we use Area Under the Receiver Operating Characteristic curve (AUROC), and Area Under the Precision-Recall Curve (AUPRC) to evaluate the binding-sites prediction performance.

## Implementation details

We trained the model by using the AdamW optimizer with Pytorch (v2.0.1), pytorch geometric (v2.3.0), and Python (v3.9.16). The experiments of protein and drug interaction predictions and protein binding site predictions were performed five times on an NVIDIA GeForce RTX 4090 GPU, and the experiments on protein contact interface predictions were performed on A800 GPU with a large GPU memory. The iteration k of the EM framework was usually converged in 5-8 iterations. The hyper-parameters on each task and additional ablation study are presented in Supplementary information B.

**Reporting summary**

Further information on research design is available in the Nature Portfolio Reporting Summary linked to this article.

## Data availability

The PPI and protein data used in this study are obtained from the previous study (HIGH-PPI), which are available in the Zenodo database under accession code https://doi.org/10.5281/zenodo.7213401. The native protein structures are obtained from PDB: https://www.rcsb.org/. The DPI data are obtained from the previous study (ConPLex), which are available on GitHub (https://github.com/samsledje/ConPLex_dev/tree/main/dataset/BIOSNAP). The DDI data are obtained from GitHub (https://github.com/isjakewong/MIRACLE/tree/main/MIRACLE/datachem). The DIPS-Plus are obtained from the previous study (DeepInteract), which is available in https://github.com/BioinfoMachineLearning/DIPS-Plus. The protein-protein binding sites dataset is obtained from https://github.com/jertubiana/ScanNet/tree/main/datasets. Source data are provided with this paper.

## Code availability

The source code of MUSE is available at https://github.com/biomed-AI/MUSE. A Zenodo version is also available at https://doi.org/10.5281/zenodo.11097139 (ref. 51).

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

## Acknowledgements

This study has been supported by the National Key R&D Program of China (2022YFF1203100), the National Natural Science Foundation of China (T2394502), and the Fundamental Research Funds for the Central Universities (Sun Yat-sen University, 22lglj08).

## Author contributions

J.R. and Y.Y. conceived and supervised the project. J.R., J.X., and S.Z. contributed to the algorithm implementation. J.R., S.Z., Y.L., and Y.Y. wrote the manuscript. J.R., Y.L., S.Z., and Y.Y. discussed and performed the rebuttal experiments. All authors were involved in the discussion and proofreading.

## Competing interests

The authors declare no competing interests.
