## [Peer Review File · Nature Communications]

Reviewers' Comments:

Reviewer #1:

Remarks to the Author:

What are the noteworthy results

The paper proposes a variational expectation-maximization framework, MUSE, for multi-scale learning of drug and protein interactions, which achieves improved results on learning tasks for each scale compared to previous multi-scale learning frameworks.

The authors tackle the issue of unbalanced greedy learning in previous multi-scale learning frameworks through mutual supervision and iterative optimization, packaged as a variant of the expectation-maximization algorithm.

Will the work be of significance to the field and related fields? How does it compare to the established literature?

Yes, the work will be of significance to the protein and drug interaction modeling communities. With improving experimental techniques and data being generated at different scales, it is important to have learning methods that can accommodate and leverage such multi-scale data.

The method addresses the main issue plaguing previous literature -- unbalanced & greedy learning in multi-scale methods. They do so via a probabilistic model that encourages mutual supervision and iterative optimization, resulting in a variant of the well studied expectation-maximization algorithm. The authors also show empirical evidence for the improved learning upon fixing this issue.

Does the work support the conclusions and claims, or is additional evidence needed?

Yes, the work supports the conclusions and claims with sufficient empirical backing. They conduct ablation studies diagnosing the issues associated with the previous multi-scale frameworks, and show how their proposed model addresses this issue.

Are there any flaws in the data analysis, interpretation and conclusions? Do these prohibit publication or require revision?

No, there are no flaws present in the data analysis and conclusions. The authors diagnose the issue of greedy and unbalanced learning in multi-scale methods by comparing the information utilization for different scales relative to a single-scale method and showcase the more balanced ratio of MUSE compared to existing multi-scale methods in HIGH-PPI and MIRACLE.

The datasets the models are trained on and evaluated are the standard ones used in the field for these tasks.

A minor corrections that should be fixed:

In Figure 2 and the associated explanation paragraph -- "Specifically, on the PPI dataset, MUSE outperformed all existing models including single-scale (DrugVQA [8] and TAG-PPI[25]), and multi-view methods (GNN-PPI [22] and HIGH-PPI [15])." -- the curve corresponding to DrugVQA cannot be found in Figure 2.

Is the methodology sound? Does the work meet the expected standards in your field?

The proposed probabilistic framework is sound, and the derivations follow logically. The choice of neural architectures are well motivated from prior literature.

Is there enough detail provided in the methods for the work to be reproduced?

Yes, the provided code is easy to follow, can be installed (with minor modifications), and the training script provided runs. The hyperparameters and other training choices are described in the

supplementary.

Reviewer #2:

Remarks to the Author:

This paper proposed a multi-scale framework combining inter-molecular and intra-molecular graph to predict the interactions between molecules. To learn the framework, the paper proposed variational EM, in which E-step is used to optimize the intra-molecular model and M-step is used to optimize the inter-molecular model. The proposed method achieves SOTA performance on several benchmarks.

Advantages:

1. Novel Method: The proposed method is novel, which integrates multi-scale information and uses variational EM to learn the proposed framework.
2. Strong Performance: The model shows strong prediction ability, achieving SOTA performance on several molecular interaction prediction benchmarks.

Disadvantages:

1. The motivation is unclear: The biggest concern to me of this paper is I don't really get the reason why the author used variational EM to learn the framework. Initially, in Section 4.3, when the author mentioned latent variables, I don't know what the latent variables refer to here. Secondly, in E-step and M-step, the author tried to respectively minimize the KL divergence between the approximate posterior and true posterior and maximize the likelihood. But according to the optimization object 11 and 14, they seem just semi-supervised learning to me with sampled labels. It seems the author can merge the two steps together and use semi-supervised learning combined data augmentation with sampled pseudo labels. I don't really see the necessity to use variational EM.
2. The claims are not well proved: Section 2.2 and 2.3 respectively claim leveraging intra/inter molecular information for improved predictions at the inter/intra-molecular scale. However, in both sections, I didn't see any ablation study, but just the performance of the proposed MUSE. I can't agree with the authors' claims.
3. The issue in derivation: KL divergence is not symmetrical. In equation 5, C denotes the entropy of p_{θ} . However, $KL(Q||P)$ is not equal to $KL(P||Q)$. Then according to equation 7, C should denote the entropy of q_{ϕ} , which can not be directly removed in E-step as it contains the related parameters ϕ .
4. Grammar error: There are also some grammar errors, such as "structures information" in 2.2 title → "structure information"

Additional questions:

1. How many iterations did the author use in EM algorithm?
2. How was the utilization rate calculated in Figure 4a?
3. Does the training steps in Figure 4b refer to the optimization step or EM iteration number?
4. Did the author try ESM2 model, which achieves SOTA performance on many protein understanding tasks such as binding site prediction or PPI prediction?

Reviewer #3:

Remarks to the Author:

This manuscript reports to describe a method for predicting interaction networks (i.e. which of a set of proteins and drugs interact with each other) and the interfaces of these molecules that interact. The method, called MUSE, is intended to take a multiscale approach that balances intra-molecular structural graphs and inter-molecular network information using an iterative expectation maximization approach.

Unfortunately, I found it impossible to understand the method or the results presented.

The abstract is too vague. In particular, it's not clear what the authors mean by inter/intra molecular and how the two aspects are connected by their model.

The first section of the results isn't very informative. As described, I can't tell if the algorithm does expectation maximization or just iterates between two different scales. What are the inputs/outputs of the expectation and maximization steps? What labels are updated during the maximization step? What's predicted during the expectation step? What information is shared at the two different scales? These questions were not adequately addressed in the Methods or other parts of the manuscript either.

Figure 1 is very hard to follow. There's no legend/key explaining the colors or what nodes/edges in the different networks represent.

No context/explanation of the datasets the method was applied to are given. There's also no discussion of the metrics used to evaluate each method. For a broad audience, one should remind the reader what an F1 score is, for example.

Reviewer #4:

Remarks to the Author:

The manuscript 'A Variational Expectation-Maximization Framework for Balanced Multi-scale Learning of Protein and Drug Interactions' submitted by Rao et al deals with a multi-scale representation learning framework based on a variational approximation of the EM algorithm in order to alleviate the problem of biased optimization to one scale in multi-scale learning, i.e., optimization imbalance. The authors integrated two scales, intra-molecular and inter-molecular scales of biomolecules, into a multi-scale framework and showed that the iterative optimization improved the multi-scale representation learning. This iterative optimization is the core of their proposed method, which can interactively capture both intra-molecular structures and inter-molecular network information, with different learning rates at the two scales. They demonstrated that their method outperforms current state-of-the-art models not only in prediction at the inter-molecular scale but also at the intra-molecular scale. In addition, they showed that their method has the potential to be extended to other multi-scale tasks. Their method is novel in that it provides a perspective on how to interactively integrate and optimally utilize data related to biomolecules at different scales. Furthermore, the authors have rigorously evaluated the robustness of their method from various viewpoints, increasing the reliability of their method. I have only minor questions and comments:

Minor questions and comments:

- Page 1, in the abstract: It is trivial, but it would be better to use "imbalanced nature" than "unbalanced nature". In general, "imbalanced" indicates that something is out of proportion (out of balance), while "unbalanced" indicates that someone or something is unstable.
- Page 3, in the section 2.1: "The EM framework optimises ..." should be "The EM framework optimizes ...".
- Page 4, in the section 2.4: "the utilization rates of MUSE for the intra- and inter- molecular scales rose to 0.103 and 0.315" should be "the utilization rates of MUSE for the intra- and inter-molecular scales rose to 0.103 and 0.318". (?)
- Page 11-12, in the section 4.1: you explain how negative drug-protein pairs are generated, but do not explain in detail how negative protein-protein pairs are generated. Please clearly explain how the negative protein-protein pairs are generated and the ratio of the number of PPIs in the SHS27K dataset to the number of those pairs generated.
- Page 11, in connection with paragraph 3 of Section 3: Native protein structures are used as multi-scale learning datasets in the proposed method. In fact, there are many proteins whose

native structures are unknown, and there are cases where structures predicted by an AI systems, such as AlphaFold, are used. Does the proposed method have the ability to handle such structures and the potential to enhance the ability to handle such structures?

Response: We are very grateful to the Editor and Reviewers for their constructive suggestions, which helped us greatly improve the quality of our manuscript. We have conducted the necessary computations/analyses and revised our manuscript accordingly. The point-by-point responses are listed as follows. All edits have been colored in red in our revised manuscript version and the revised supplementary information.

Reviewer #1 (Remarks to the Author):

The paper proposes a variational expectation-maximization framework, MUSE, for multi-scale learning of drug and protein interactions, which achieves improved results on learning tasks for each scale compared to previous multi-scale learning frameworks.

Response to Reviewer#1:

We thank the reviewer for the time taken to review our work and the constructive feedback provided. We are glad that the reviewer gave positive feedback on our paper, especially on our novel methodology and sufficient empirical experiments. We have addressed the specific minor questions raised by the reviewer below.

A minor correction that should be fixed:

1. In Figure 2 and the associated explanation paragraph -- “Specifically, on

the PPI dataset, MUSE outperformed all existing models including single-scale (DrugVQA¹ and TAG-PPI²), and multi-view methods (GNN-PPI³ and HIGH-PPI⁴). “ -- the curve corresponding to DrugVQA cannot be found in Figure 2.

Response: Thanks for the suggestions. We have added the curve corresponding to DrugVQA in Figure 2 and revised our final version of the manuscript.

2. The provided code is easy to follow, can be installed (with minor modifications), and the training script provided runs. The hyperparameters and other training choices are described in the supplementary.

Installation on the Linux machine failed with the provided command, but the environment could be set up by installing the major packages as listed in the README + a few extra ones (mol2vec, genism, pandas, scipy, omegaconf, descriptastorus)

The training script runs from the provided config and instructions. It would be nice if the authors could add CPU running support, but this is not a requirement.

Response: We thank the reviewer for pointing this out. We have revised the installation details and added the supported CPU environments. We also added the description to the README file on our GitHub page: <https://github.com/biomed-AI/MUSE>.

"We provide a script to install the environment, including all the necessary packages (mol2vec, gensim, pandas, scipy, omegaconf, descriptastorus). Both CPU and CUDA11.8 environments are supported."

Reviewer #2 (Remarks to the Author):

This paper proposed a multi-scale framework combining inter-molecular and intra-molecular graph to predict the interactions between molecules. To learn the framework, the paper proposed variational EM, in which E-step is used to optimize the intra-molecular model and M-step is used to optimize the inter-molecular model. The proposed method achieves SOTA performance on several benchmarks.

Advantages:

1. Novel Method: The proposed method is novel, which integrates multi-scale information and uses variational EM to learn the proposed framework.
2. Strong Performance: The model shows strong prediction ability, achieving SOTA performance on several molecular interaction prediction benchmarks.

Response to Reviewer #2:

We appreciate the reviewer for reviewing our paper carefully and giving valuable suggestions and constructive comments. We have addressed the reviewer's comments and revised our manuscript accordingly.

Disadvantages:

1. The motivation is unclear: The biggest concern to me of this paper is I don't really get the reason why the author used variational EM to learn the framework. Initially, in Section 4.3, when the author mentioned latent variables, I don't know what the latent variables refer to here. Secondly, in E-step and M-step, the author tried to respectively minimize the KL divergence between the approximate posterior and true posterior and maximize the likelihood. But according to the optimization object 11 and 14, they seem just semi-supervised learning to me with sampled labels. It seems the author can merge the two steps together and used semi-supervised learning combined data augmentation with sampled pseudo labels. I don't really see the necessity to use variational EM.

Response: Thanks for your valuable comments. Here, the latent variables are the learned variables of objects from models, i.e., the interaction labels in our task. We have added an equation, revised this statement in E-step and M-step, and changed the flow chart in Fig 1 as below to make it clear.

"Our framework tries to maximize the log-likelihood function of the observed interaction labels, i.e. $\log p_{\theta}(y_{\mathcal{L}_V}|\mathcal{G}, A)$. It is computationally intractable to compute this log-likelihood as it requires integration over all combinations of object labels, i.e. $\log \prod_{l \in \mathcal{L}_V} p_{\theta}(y_l|\mathcal{G}, A)$."

As shown in the plot below, the vanilla model uses the merging of two single-scale models together, as used in the previous studies, such as HIGH-PPI⁴ and ScanNet⁵. These training strategies tend to rely intensively on a single scale and under-fitting the others, attributed to the imbalanced nature and inherent greediness of multi-scale learning, as evidenced by Section 2.4. Therefore, it is necessary to propose our EM framework to alternately optimize two modules (E-step and M-step) with different scales, ensuring the balanced utilization of effective information and achieving promising improvements at different scales.

(Introduction) "An intuitive approach for learning multi-scale representations is to combine the molecular graph with an interaction network and optimize them jointly.

However, due to the imbalanced nature and inherent greediness of multi-scale learning, these models often intensively rely on a single scale and under-fitting the others. This imbalanced nature prevents these approaches from effectively leveraging all informative scale-related information and often results in worse generalization."

Specifically, we changed the statement "inter-molecular scale" to "Molecular Network Scale", representing the scale of the molecule-level network between biomolecules, and the "intra-molecular scale" to "Atomic Structure Scale", representing the scale of the atom-level structure of

biomolecules.

Figure 1 (Revised). (A) The vanilla approach for learning multi-scale representations is to combine the two single-scale models and optimize them jointly. (B) The EM framework (MUSE) for Multi-scale Learning. The Expectation step trains a model with structural information of protein and drug as input to fit the known and pseudo interactions (the predicted interactions from the M-step except the first iteration). The updated interactions and structural embeddings were input to the M-step, where the molecular network was constructed to maximize the known interactions and the predicted interactions from the E-step. The updated interactions were sent to E-step for new iterations.

2. The claims are not well proved: Section 2.2 and 2.3 respectively claims leveraging intra/inter molecular information for improved predictions at the inter/intra-molecular scale. However, in both sections, I didn't see any ablation study, but just the performance of the proposed MUSE. I can't agree with the authors' claims.

Response: Thanks for your valuable comments. According to your suggestions, we have revised our manuscript and provided additional experiment results. Specifically, we add the ablation study of MUSE-Joint, which integrates two scale models and optimizes them jointly with multiple iterations, in Section 2.2 and Section 2.3 (Figure 2C and Figure 3C), demonstrating leveraging intra/inter-molecular information for improved predictions at the atomic structural and the molecular network scale. The MUSE-Joint is also slightly better than HIGH-PPI, as HIGH-PPI does not optimize the structural information jointly for interaction predictions. More ablation studies on different modules of MUSE (such as iterative optimization module, mutual supervision module, and different scale module) have been discussed in Section 2.4 (Figure 4b-c) and Section 2.5 (Figure 5c).

Figure 2 (C) (Revised). Barplot shows the best micro-F1 scores (Best-F1) or ROC-AUC of baseline, ablation study MUSE-Joint and MUSE predictions respectively on PPI predictions (Random, BFS, and DFS splits).

Figure 3 (C) (Revised). MUSE achieves state-of-the-art performance on the Masif-site test sets of PPBS tasks compared with state-of-the-art baselines and an ablation study MUSE-Joint.

(Section 2.2) "Furthermore, our model also showed improvements over the ablation study MUSE-Joint, which integrates two scale models and optimizes them jointly, attributed to the efficient utilization of the EM

framework. The MUSE-Joint is also slightly better than HIGH-PPI, as HIGH-PPI does not optimize the structural information jointly for interaction predictions."

(Section 2.3) "MUSE also performed best on another benchmark dataset developed by Masif-site, 2.23% and 1.40% better than PeSTo and MUSE-Joint respectively. "

(Section 2.4) "The continuous increase in accuracy (red curve) shows how MUSE continuously alleviates the imbalance characteristics of multi-scale learning and improves its multi-scale utilization. The learned representations by MUSE clearly classify the interaction type between protein and protein while both molecular network model (MUSE without atomic structure scale learning) and MUSE-Joint (MUSE which integrates two scale models and optimizes them jointly) have a small number of samples mixed together. (Fig. 4c)"

3. The issue in derivation: KL divergence is not symmetrical. In equation 5, C denotes the entropy of p_{θ} . However, $KL(Q||P)$ is not equal to $KL(P||Q)$. Then according to equation 7, C should denote the entropy of q_{ϕ} , which cannot be directly removed in E-step as it contains the related parameters ϕ .

Response: Thanks for the comment. To avoid directly optimizing the KL divergence in Eq 5, we followed the idea of GMNN⁶ to utilize the wake-

sleep algorithm (Hinton et al.⁷) to minimize the reverse KL divergence and C denotes as const number. We have added the detailed proof in our revised supplementary information.

(Method) "Recall that the goal of Eq. 5 for q_ϕ is to minimize the KL divergence between $q_\phi(y_{\mathcal{L}_U}|\mathcal{G}, \mathcal{D})$ and $p_\theta(y_{\mathcal{L}_U}|y_{\mathcal{L}_V}, \mathcal{G}, A)$, we followed the idea of GLEM and GMNN to utilize the wake-sleep algorithm for minimizing the reverse KL divergence (refer to Supplementary information A for detailed proofs):

$$\begin{aligned}
\mathcal{O}(q_\phi(y_{\mathcal{L}_U})) &= -KL(q_\phi(y_{\mathcal{L}_U}|\mathcal{G}, \mathcal{D}) \| p_\theta(y_{\mathcal{L}_U}|y_{\mathcal{L}_V}, \mathcal{G}, A)) \\
&= \sum_{y_{\mathcal{L}_U}} q_\phi(y_{\mathcal{L}_U}|\mathcal{G}, \mathcal{D}) [\log p_\theta(y_{\mathcal{L}_U}|y_{\mathcal{L}_V}, \mathcal{G}, A) - \log q_\phi(y_{\mathcal{L}_U}|\mathcal{G}, \mathcal{D})] \\
&= \sum_{y_{\mathcal{L}_U}} \left(\prod_{L_{ij}} q_\phi(y_{L_{ij}}|\mathcal{G}, \mathcal{D}) \right) [\log p_\theta(y_{\mathcal{L}_U}|y_{\mathcal{L}_V}, \mathcal{G}, A) - \sum_{L_{ij}} \log q_\phi(y_{L_{ij}}|\mathcal{G}, \mathcal{D})] + const \\
&= \sum_{y_{L_{ij}}} \sum_{y_{\mathcal{L}_U \setminus L_{ij}}} (q_\phi(y_{L_{ij}}) \prod_{L' \neq L_{ij}} q_\phi(y_{L'})) [\log p_\theta(y_{\mathcal{L}_U}|y_{\mathcal{L}_V}, \mathcal{G}, A) - \sum_{L'} \log q_\phi(y_{L'}|\mathcal{G}, \mathcal{D})] + const \\
&= \sum_{y_{L'}} \log \mathcal{F}(y_{L'}|\mathcal{G}, \mathcal{D}) - \sum_{y_{L'}} \log q_\phi(y_{L'}|\mathcal{G}, \mathcal{D}) + const \\
&= -KL(q_\phi(y_{\mathcal{L}'}|\mathcal{G}, \mathcal{D}) \| \frac{\mathcal{F}(y_{\mathcal{L}'})}{Z}) + const
\end{aligned}$$

where Z is a normalization term and $\mathcal{F}(y_{\mathcal{L}'})$ is the distribution on $y_{\mathcal{L}'}$:

$$\mathcal{F}(y_{\mathcal{L}'}) = \mathbb{E}_{q_\phi(y_{\mathcal{L}_U \setminus L_{ij}}|\mathcal{G}, \mathcal{D})} [\log p_\theta(y_{\mathcal{L}_U}|y_{\mathcal{L}_V}, \mathcal{G}, A)]$$

4. Grammar error: There are also some grammar errors, such as “structures information” in 2.2 title → “structure information”

Response: Thanks for the suggestions. The statement has been revised.

5. Additional questions:

(1). How many iterations did the author use in EM algorithm?

Response: Thanks for the comment. We have listed the hyperparameters for each task in the supplementary information (Section B) and the hyperparameter iteration k of the EM framework usually converged in the 5-8 iterations, depending on the task. We have added the descriptions in our revised main manuscript (Section 4.13).

(Section 4.13) "The iteration k of the EM framework was usually converged in 5-8 iterations. The hyper-parameters on each task are presented in the supplementary information B."

(2). How was the utilization rate calculated in Figure 4a?

Response: Thanks for the comment. The calculation of the utilization rate in Figure 4a have been described in the supplementary information. We have moved the descriptions in our revised main manuscript (Section 4.11).

(Section 4.11) "For a multi-scale model M , taking two scales s_0 (atomic structure scale) and s_1 (molecular network scale) as inputs, its utilization rates for s_0 and s_1 are defined as:

$$u(s_0|s) = \frac{f(M) - f(M_1)}{f(M_1)},$$

and

$$u(s_1|s) = \frac{f(M) - f(M_0)}{f(M_0)}$$

Here, M_0 and M_1 represent single-scale models obtained from the multi-scale model M , while f denotes the predicted evaluation function.

The utilization rate is the relative change in accuracy between the two models within each pair. For example, $u(s_0|s)$ measures the incremental impact of the atomic structure scale s_0 on improving the accuracy of label predictions. It is constrained within the range of -1 to 1. A higher utilization rate indicates that the multi-scale model can benefit from incorporating this scale."

(3). Do the training steps in **Figure 4b** refer to the optimization step or EM iteration number?

Response: Thanks for the comment. The training steps in Figure 4b are the EM iteration step for MUSE and for MUSE-Joint. We have revised the presentation carefully and provided additional descriptions in the caption of Figure 4b as below.

Figure 4b (Revised). The convergence curves of MUSE and MUSE-Joint on the PPI dataset during the EM optimization iteration steps. The red curve in iterative optimization shows how MUSE continuously alleviates the imbalance characteristics of multi-scale learning and improves its multi-scale utilization.

(4). Did the author try ESM2 model, which achieves SOTA performance on many protein understanding tasks such as binding site prediction or PPI prediction?

Response: Thanks for your suggestions. We have added the comparison of our basic GNN model with the ESM2 model in our MUSE framework for PPI prediction. As shown in the Table below, at the structural scale, the inclusion of ESM2 (MUSE-structure w/ ESM2) could significantly improve PPI predictions over MUSE-structure (w/ GNN). At the network scale, ESM2 led to improved performance but the improvement is smaller. When fusing the structural and the network scale, ESM2 didn't bring improvements in performance but converged faster (converged in 5 iterations).

Response-Table 1

	AUROC	Best-F1
MUSE-structure (w/ GNN)	0.904	0.781
MUSE-structure (w/ ESM2)	0.927	0.831
MUSE-network (w/ GNN embeddings)	0.939	0.921
MUSE-network (w/ ESM2 embeddings)	0.943	0.930
MUSE (w/ GNN)	0.971	0.952 (in 8-iter)
MUSE (w/ ESM2)	0.974	0.950 (in 5-iter)

(Supplementary Information C.2) "As shown in Table S5, at the structural scale, the inclusion of ESM2 (MUSE-structure w/ ESM2) could significantly improve PPI predictions over MUSE-structure (w/ GNN). At the network scale, ESM2 led to improved performance but the improvement is smaller. When fusing the structural and the network scale, ESM2 didn't bring improvements in performance but converged faster (converged in 5 iterations)."

Reviewer #3 (Remarks to the Author):

This manuscript proports to describe a method for predicting interaction networks (i.e. which of a set of proteins and drugs interact with each other) and the interfaces of these molecules that interact. The method, called MUSE, is intended to take a multiscale approach that balances intra-

molecular structural graphs and inter-molecular network information using an iterative expectation maximization approach.

Response to Reviewer #3:

We thank the reviewer for the enthusiastic comments on our paper, which have helped us further improve the quality of our manuscript. We have addressed the reviewer's specific questions below.

Unfortunately, I found it impossible to understand the method or the results presented.

1. The abstract is too vague. In particular, it's not clear what the authors mean by inter/intra molecular and how the two aspects are connected by their model.

Response: Thanks for your valuable comments. To make it clear we have carefully revised our abstract. Specifically, we changed the statement "inter-molecular scale" to "Molecular Network Scale", representing the scale of the molecule-level network between biomolecules, and the "intra-molecular scale" to "Atomic Structure Scale", representing the scale of the atom-level structure of biomolecules.

(Abstract) "While a few multi-view learning methods are devoted to fusing the multi-scale information, these methods tend to rely intensively on a single scale and under-fitting the others, likely attributed to the imbalanced nature and inherent greediness of multi-scale learning. To

alleviate the optimization imbalance, we present MUSE, a multi-scale representation learning framework based on a variant expectation maximization to optimize different scales in an alternating procedure over multiple iterations. This strategy efficiently fuses multi-scale information between atomic structure and molecular network through mutual supervision and iterative optimization. MUSE outperforms the current state-of-the-art models not only in molecular interaction (protein-protein, drug-protein, and drug-drug) tasks but also in protein interfaces prediction at the atomic structure scale."

2. The first section of the results isn't very informative. As described, I can't tell if the algorithm does expectation maximization or just iterates between two different scales. What are the inputs/outputs of the expectation and maximization steps? What labels are updated during the maximization step? What's predicted during the expectation step? What information is shared at the two different scales? These questions were not adequately addressed in the Methods or other parts of the manuscript either.

Response: Thanks for your valuable comments. We have revised the explanations of our algorithm procedure carefully. Our method iteratively optimizes two modules (E-step and M-step) at two different scales. The E-step takes the pair of protein and drug with their atom-level structural information as input, augments with the predicted interactions from the M-

step, and outputs with the predicted pairs. The M-step takes the molecule-level interaction network and the structural embeddings and predicted interactions from the E-step as the input and also outputs the predicted interactions. The shared information between the two different scales is the predicted unobserved labels. We have added the explanations in the updated manuscript. The algorithm procedure is shown in the revised Figure 1.

(Section 2.1) "The E-step takes the pair of protein and drug with their atom-level structural information as input, augments with the predicted interactions from the M-step (except for the first iteration), and outputs with the predicted pairs. The M-step takes the molecule-level interaction network and the structural embeddings and predicted interactions from the E-step as the input and also outputs the predicted interactions. "

(E-step Objective function) "The final objective function for training the GNN_ϕ model of atomic structure scale:

$$\begin{aligned} \mathcal{L}_{structure} &= \mathcal{O}_{q_\phi} \\ &= \alpha \sum_{L_{ij} \in \mathcal{L}_U} \mathbb{E}_{p(y_{L_{ij}} | y_{\mathcal{L}_V}, \mathcal{G}, A, \hat{y}_{\mathcal{L}_U \setminus L_{ij}})} [\log q(y_{L_{ij}} | \mathcal{G}, \mathcal{D})] + (1 \\ &\quad - \alpha) \sum_{L_{ij} \in \mathcal{L}_V} \log q(y_{L_{ij}} | \mathcal{G}, \mathcal{D}) \end{aligned}$$

Intuitively, the first term could be viewed as a knowledge-distilling process

that teaches the model of atomic structure scale GNN_ϕ by forcing it to predict the label distribution based on the predicted pseudo interaction from the molecular network model GNN_θ . The second term is a supervised objective which uses the given labeled interactions for training."

(M-step Objective function) "The final objective function for training the GNN_θ model of molecular network scale:

$$\begin{aligned} \mathcal{L}_{network} &= \mathcal{O}_\theta \\ &= \beta \sum_{L_{ij} \in \mathcal{L}_U} \log p_\theta(\hat{y}_{L_{ij}} | \mathcal{G}, A, y_{\mathcal{L}_V}, \hat{y}_{\mathcal{L}_U \setminus L_{ij}}) + (1 \\ &\quad - \beta) \sum_{L_{ij} \in \mathcal{L}_V} \log p_\theta(y_{L_{ij}} | \mathcal{G}, A, y_{\mathcal{L}_V \setminus L_{ij}}, \hat{y}_{\mathcal{L}_U}), \end{aligned}$$

where β is a hyperparameter that is added to balance the weight of the two terms. The first term can be viewed as a knowledge distillation process that injects the knowledge (predicted interactions) captured by the model of atomic structure scale GNN_ϕ into the molecular network scale model GNN_θ via all the pseudo-labels. The second term is simply a supervised loss, where we use observed interaction labels for model training.

Figure 1 (Revised). (A) The vanilla approach for learning multi-scale representations is to combine the two single-scale models and optimize them jointly. (B) The EM framework (MUSE) for Multi-scale Learning. The Expectation step trains a model with structural information of protein and drug as input to fit the known and pseudo interactions (the predicted interactions from the M-step except the first iteration). The updated interactions and structural embeddings were input to the M-step, where the molecular network was constructed to maximize the known interactions and the predicted interactions from the E-step. The updated interactions were sent to E-step for new iterations.

3. Figure 1 is very hard to follow. There's no legend/key explaining the colors or what nodes/edges in the different networks represent.

Response: To make our contributions clear, we have reformulated the workflow of Figure 1, as shown above.

4. No context/explanation of the datasets the method was applied to are given. There's also no discussion of the metrics used to evaluate each method. For a broad audience, one should remind the reader what an F1 score is, for example.

Response: Thanks for your comments. We have revised the explanation of those datasets in Section 4.1 and the metrics used to evaluate each method also have been added in Section 4.12.

(Datasets Section 4.1) We have evaluated our framework on three multi-scale interaction prediction benchmarks, the protein inter-chain benchmark (DIPS-Plus) and the protein binding sites benchmark (Scannet). The statistics of these datasets are presented in Supplementary Information B.

(Metrics for Interaction Predictions) "For the three multi-scale interaction prediction benchmarks, i.e. SHS27K, BioSNAP, and DeepDDI, we evaluated their performance using micro-F1 (Best-F1), Area Under the Receiver Operating Characteristic curve (AUROC) and Area Under Precision-Recall Curve (AUPRC). In particular, for the multi-label PPI prediction, the different PPI types in the datasets we used are very

imbalanced, so micro-F1 may be preferred."

(Metrics for protein inter-chain contact Predictions) "To evaluate the prediction of protein inter-chain contact, we follow the setting established by Scannet, where a positive label is assigned to each inter-chain residue pair found within 6Å of each other in the complex's bound (i.e., structurally-conformed) state. We use Average top-k precision ($P@k$), Recall ($R@k$), and Area Under the Receiver Operating Characteristic curve (AUROC) to assess our method in predicting protein-protein interactions and the quality of the generated docking results."

(Metrics for protein binding-sites Predictions) "Following the previous studies, we use Area Under the Receiver Operating Characteristic curve (AUROC), and Area Under the Precision-Recall Curve (AUPRC) to evaluate the binding-sites prediction performance."

Reviewer #4 (Remarks to the Author):

The manuscript 'A Variational Expectation-Maximization Framework for Balanced Multi-scale Learning of Protein and Drug Interactions' submitted by Rao et al deals with a multi-scale representation learning framework based on a variational approximation of the EM algorithm in order to alleviate the problem of biased optimization to one scale in multi-scale learning, i.e., optimization imbalance. The authors integrated two scales,

intra-molecular and inter-molecular scales of biomolecules, into a multi-scale framework and showed that the iterative optimization improved the multi-scale representation learning. This iterative optimization is the core of their proposed method, which can interactively capture both intra-molecular structures and inter-molecular network information, with different learning rates at the two scales. They demonstrated that their method outperforms current state-of-the-art models not only in prediction at the inter-molecular scale but also at the intra-molecular scale. In addition, they showed that their method has the potential to be extended to other multi-scale tasks. Their method is novel in that it provides a perspective on how to interactively integrate and optimally utilize data related to biomolecules at different scales. Furthermore, the authors have rigorously evaluated the robustness of their method from various viewpoints, increasing the reliability of their method. I have only minor questions and comments:

Response to Reviewer #4:

We thank the reviewer for the time taken to carefully review our work and the positive feedback and constructive comments provided. We are glad that the reviewer gave positive feedback on our paper, especially on the novelty of our method. We have addressed the reviewer's specific questions below.

Minor questions and comments:

1. Page 1, in the abstract: It is trivial, but it would be better to use "imbalanced nature" than "unbalanced nature". In general, "imbalanced" indicates that something is out of proportion (out of balance), while "unbalanced" indicates that someone or something is unstable.

Response: Thanks for your suggestion. We have revised the statement in our final version of the manuscript.

2. Page 3, in the section 2.1: "The EM framework optimises ..." should be "The EM framework optimizes ...".

Response: Thanks for your suggestion. We have revised the statement in our final version of the manuscript.

3. Page 4, in the section 2.4: "the utilization rates of MUSE for the intra- and inter- molecular scales rose to 0.103 and 0.315" should be "the utilization rates of MUSE for the intra- and inter- molecular scales rose to 0.103 and 0.318". (?)

Response: Thanks for your suggestion. We have revised the statement in our final version of the manuscript.

4. Page 11-12, in the section 4.1: you explain how negative drug-protein pairs are generated, but do not explain in detail how negative protein-

protein pairs are generated. Please clearly explain how the negative protein-protein pairs are generated and the ratio of the number of PPIs in the SHS27K dataset to the number of those pairs generated.

Response: Thanks for your suggestions. We have revised the explanation of protein-protein interaction datasets in Section 4.1.

(Section 4.1) "The SHS27K PPI dataset contains SHS27k (sub-dataset of STRING) with 6660 protein-protein pairs (PPIs) and 1533 human proteins with native protein structures. These PPIs are divided into 7 types, namely reaction, binding, post-translational modifications (ptmod) activation, inhibition, catalysis, and expression, which contain 15,056 positive interaction types and 31,564 negative interaction types."

5. Page 11, in connection with paragraph 3 of Section 3: Native protein structures are used as multi-scale learning datasets in the proposed method. In fact, there are many proteins whose native structures are unknown, and there are cases where structures predicted by an AI systems, such as AlphaFold, are used. Does the proposed method have the ability to handle such structures and the potential to enhance the ability to handle such structures?

Response: Thanks for your valuable comments. We have implemented the predicted structure from AlphaFold⁸ in the PPI predictions. Response-Table 2 shows the experiment results on the PPI dataset and we have

included them in supplementary information. Experiments demonstrated that our model is not significantly affected by structure errors where powerful structures are not available. We have added the experiments and discussion in our revised manuscript.

Response-Table 2

	AUROC	Best-F1
HIGH-PPI (AlphaFold)	0.903	0.806
HIGH-PPI (PDB)	0.956	0.884
MUSE (AlphaFold)	0.964	0.940
MUSE (PDB)	0.971	0.952

(Supplementary Information C.3) "As the experimental structures are not always available in real-world scenarios, we also investigated the impact on performance when using predicted structures as input for testing. As expected, the performance of HIGH-PPI and MUSE decreases, because they were trained with high-quality native structures. For example, the Best-F1 of HIGH-PPI for predicting PPI decreases from 0.884 to 0.806, compared to the Best-F1 of 0.940 by MUSE. Therefore, in the practical situations where experimental structures are unavailable, our proposed method still significantly outperforms the baseline methods."

Reference:

1. Zheng S, Li Y, Chen S, Xu J, Yang Y. Predicting drug–protein interaction using quasi-visual question answering system. *Nature Machine Intelligence* **2**, 134-140 (2020).
2. Song B, Luo X, Luo X, Liu Y, Niu Z, Zeng X. Learning spatial structures of proteins improves protein–protein interaction prediction. *Briefings in bioinformatics* **23**, bbab558 (2022).
3. Zrimšek U. Learning Unknown from Correlations: Graph Neural Network for Inter-novel-protein Interaction Prediction. In: *ML Reproducibility Challenge 2021 (Fall Edition)* (2022).
4. Gao Z, *et al.* Hierarchical graph learning for protein–protein interaction. *Nature Communications* **14**, 1093 (2023).
5. Tubiana J, Schneidman-Duhovny D, Wolfson HJ. ScanNet: an interpretable geometric deep learning model for structure-based protein binding site prediction. *Nature Methods* **19**, 730-739 (2022).
6. Qu M, Bengio Y, Tang J. Gmn: Graph markov neural networks.). PMLR.
7. Hinton GE, Dayan P, Frey BJ, Neal RM. The "wake-sleep" algorithm for unsupervised neural networks. *Science* **268**, 1158-1161 (1995).
8. Jumper J, *et al.* Highly accurate protein structure prediction with AlphaFold. *Nature* **596**, 583-589 (2021).

Reviewers' Comments:

Reviewer #1:

Remarks to the Author:

My questions and comments from the reviewing phase has been addressed by the authors now. The review is copied below for ease:

What are the noteworthy results

The paper proposes a variational expectation-maximization framework, MUSE, for multi-scale learning of drug and protein interactions, which achieves improved results on learning tasks for each scale compared to previous multi-scale learning frameworks.

The authors tackle the issue of unbalanced greedy learning in previous multi-scale learning frameworks through mutual supervision and iterative optimization, packaged as a variant of the expectation-maximization algorithm.

Will the work be of significance to the field and related fields? How does it compare to the established literature?

Yes, the work will be of significance to the protein and drug interaction modeling communities. With improving experimental techniques and data being generated at different scales, it is important to have learning methods that can accommodate and leverage such multi-scale data.

The method addresses the main issue plaguing previous literature -- unbalanced & greedy learning in multi-scale methods. They do so via a probabilistic model that encourages mutual supervision and iterative optimization, resulting in a variant of the well studied expectation-maximization algorithm. The authors also show empirical evidence for the improved learning upon fixing this issue.

Does the work support the conclusions and claims, or is additional evidence needed?

Yes, the work supports the conclusions and claims with sufficient empirical backing. They conduct ablation studies diagnosing the issues associated with the previous multi-scale frameworks, and show how their proposed model addresses this issue.

Are there any flaws in the data analysis, interpretation and conclusions? Do these prohibit publication or require revision?

No, there are no flaws present in the data analysis and conclusions. The authors diagnose the issue of greedy and unbalanced learning in multi-scale methods by comparing the information utilization for different scales relative to a single-scale method and showcase the more balanced ratio of MUSE compared to existing multi-scale methods in HIGH-PPI and MIRACLE.

The datasets the models are trained on and evaluated are the standard ones used in the field for these tasks.

A minor corrections that should be fixed:

In Figure 2 and the associated explanation paragraph -- "Specifically, on the PPI dataset, MUSE outperformed all existing models including single-scale (DrugVQA [8] and TAG-PPI[25]), and multi-view methods (GNN-PPI [22] and HIGH-PPI [15])." -- the curve corresponding to DrugVQA cannot be found in Figure 2.

Is the methodology sound? Does the work meet the expected standards in your field?

The proposed probabilistic framework is sound, and the derivations follow logically. The choice of neural architectures are well motivated from prior literature.

Is there enough detail provided in the methods for the work to be reproduced?

Yes, the provided code is easy to follow, can be installed (with minor modifications), and the training script provided runs. The hyperparameters and other training choices are described in the supplementary.

Reviewer #2:

Remarks to the Author:

I have carefully read the responses from the authors, and I think most of my concerns have been addressed. I have two follow-up questions:

1. "Therefore, it is necessary to propose our EM framework to alternately optimize two modules (E-step and M-step) with different scales, ensuring the balanced utilization of effective information and achieving promising improvements at different scales"

In EM algorithm, E-step is used to calculate the expectation based on the current approximate posterior, and M-step maximizes the expectation computed in E-step to optimize the parameter. I can't understand why the two steps are necessary to "ensure the balanced utilization of effective information and achieving promising improvements at different scales". Could the author provide more insights into the relationship between molecule/atomic and E-step/M-step?

2. "We add the ablation study of MUSE-Joint, which integrates two scale models and optimizes them jointly with multiple iterations"

The authors changed "inter-molecular scale" to "Molecular Network Scale", representing the scale of the molecule-level network between biomolecules, and the "intra-molecular scale" to "Atomic Structure Scale". This is good, making the paper easier to understand. I also appreciate the authors' effort in providing the experimental results of MUSE-joint. Essentially, what I would like to see is: as the author claimed "leveraging intra/inter molecular information for improved predictions at the inter/intra-molecular scale", how the author can prove both molecule and atomic networks are beneficial for improved performance instead of only using one of them? The author didn't provide any this ablation study using the proposed framework.

Reviewer #3:

Remarks to the Author:

I'm still unclear what this manuscript contributes.

Reviewer #4:

Remarks to the Author:

The authors have sincerely addressed my concerns in a satisfactory manner. There is no need for me to review their revised manuscript again. Therefore, unless the other reviewer finds any concerns in the revised version, I recommend that their work be published.

Reviewer #2 (Remarks to the Author):

I have carefully read the responses from the authors, and I think most of my concerns have been addressed. I have two follow-up questions:

1. In EM algorithm, E-step is used to calculate the expectation based on the current approximate posterior, and M-step maximizes the expectation computed in E-step to optimize the parameter. I can't understand why the two steps are necessary to "ensure the balanced utilization of effective information and achieving promising improvements at different scales". Could the author provide more insights into the relationship between molecule/atomic and E-step/M-step?

Response: Thanks for your valuable comments.

According to the greedy learner hypothesis [1,2], the multi-scale model has different learning speeds at different scales, which change continuously during training. The vanilla multi-scale model, which directly combines two single-scale models, may cause the neural network to prioritize the most informative scale, allowing it to learn faster. However, this leads the model to local optima and limits its use of information from other scales. It is difficult to dynamically adjust the learning speed of different scales during the training process [1]. Therefore, we proposed the EM framework (MUSE) to learn specific information at each scale with different learning rates by alternately optimizing E-step and M-step to ensure balanced utilization of effective information.

In the E-step, the goal is to use the atomic structural scale model to update the variational distribution $q(y_{\mathcal{L}_U} | y_{\mathcal{L}_V}, \mathcal{G}, \mathcal{D})$ to approximate the true posterior distribution $p(y_{\mathcal{L}_U} | y_{\mathcal{L}_V}, \mathcal{G}, A)$. Thus, we estimated the expectation for $q(y_{\mathcal{L}_U})$ based on the structure representation from the atomic structure scale model. In the M-step, we used the structural representations as node features and the estimated interactions as pseudo-labels to further maximize the expectations of the unknown interactions $\mathbb{E}_{q_\phi(y_{\mathcal{L}_U} | \mathcal{G}, \mathcal{D})} [p_\theta(y_{\mathcal{L}_U}, y_{\mathcal{L}_V} | \mathcal{G}, A)]$ and make accurate predictions with the molecular network scale model.

$\mathcal{G} := \{G_i\}_{i=1}^N$ is the set of biomolecules, the known interactions are denoted as the pairs of molecular atomic structural graphs $\mathcal{D} = \{(G_i, G_j)\}_{(i,j)}^M$, A represents the adjacency matrix in the network, \mathcal{L}_U is the set of unknown interactions, and $y_{\mathcal{L}_V}$ is the labels of the observed interactions \mathcal{L}_V .

"The lower bound in $\log p_\theta(y_{\mathcal{L}_V} | \mathcal{G}, A)$ (Eq. 1) can be optimized through the iterative process of alternating between the variational E-step and the M-step. In the variational

E-step, the goal is to fix p_θ and update q_ϕ to minimize the KL divergence between the variational distribution $q_\phi(y_{\mathcal{L}_U}|\mathcal{G}, \mathcal{D})$ and the true posterior distribution $p_\theta(y_{\mathcal{L}_U}|y_{\mathcal{L}_V}, \mathcal{G}, A)$. In the M-step, we aim to update p_θ to maximize the below function:

$$\mathcal{O}(\theta) = \mathbb{E}_{q_\phi(y_{\mathcal{L}_U}|\mathcal{G}, \mathcal{D})} [p_\theta(y_{\mathcal{L}_U}, y_{\mathcal{L}_V}|\mathcal{G}, A)]$$

[1] Wu, N., Jastrzebski, S., Cho, K., Geras, K.J.: Characterizing and overcoming the greedy nature of learning in multi-modal deep neural networks. In: International Conference on Machine Learning, pp. 24043–24055 (2022). PMLR

[2] Fan, Y., Xu, W., Wang, H., Wang, J., Guo, S.: Pmr: Prototypical modal rebalance for multimodal learning. In: Proceedings of the IEEE/CVF Conference on Computer Vision and Pattern Recognition, pp. 20029–20038 (2023)

2. The authors changed "inter-molecular scale" to "Molecular Network Scale", representing the scale of the molecule-level network between biomolecules, and the "intra-molecular scale" to "Atomic Structure Scale". This is good, making the paper easier to understand. I also appreciate the authors' effort in providing the experimental results of MUSE-joint. Essentially, what I would like to see is: as the author claimed "leveraging intra/inter molecular information for improved predictions at the inter/intra-molecular scale", how the author can prove both molecule and atomic networks are beneficial for improved performance instead of only using one of them? The author didn't provide any this ablation study using the proposed framework.

Response: Thanks for your valuable comments. We have included these ablation studies in the last version of the manuscript (Section 2.4, Supplementary Section C.2), but didn't put them together. To make it clear, we have organized the ablation experiments in our final revised manuscript. Specifically, we have provided the ablation study of the MUSE-structure, MUSE-network, and MUSE-joint in the protein-protein interaction prediction task.

MUSE-structure, which solely focuses on the atomic structure scale, achieved the poorest performance. The MUSE-network outperformed MUSE-structure, demonstrating that the natural information imbalance of different scales exists in the current dataset. MUSE-joint, which integrates two scale models and optimizes them jointly with multiple iterations, showed substantial improvements over the MUSE-structure and MUSE-network. Furthermore, our model MUSE also showed improvements over the ablation study MUSE-joint because of its efficient utilization for multi-scale learning with the proposed EM framework.

Response Table 1. The ablation study of MUSE.

	AUROC	Best-F1
MUSE-structure	0.904	0.781
MUSE-network	0.932	0.874
MUSE-joint	0.962	0.929
MUSE	0.971	0.953

Reviewer #3 (Remarks to the Author):

I'm still unclear what this manuscript contributes.

Response:

Thanks for your comments. We have revised our manuscript to make our contribution clearer.

Our contributions can be summarized as:

1. We present MUSE, a multi-scale representation learning framework based on a variant expectation maximization, which can effectively integrate multi-scale information for learning.
2. The framework effectively addresses the optimization imbalance in multi-scale learning through mutual supervision and iterative optimization, allowing MUSE to be extended to other multi-scale tasks.
3. Extensive experiments show that MUSE outperforms the state-of-the-art methods not only in predicting molecular interactions but also in predicting molecular binding interfaces.

"In this study, we present MUSE, a multi-scale representation learning framework based on a variant expectation maximization, which can effectively integrate multi-scale information for learning. In contrast to existing methods that rely heavily on single-scale information, MUSE effectively addresses the optimization imbalance in multi-scale learning through mutual supervision and iterative optimization. Extensive experiments show that MUSE outperforms the state-of-the-art methods not only in predicting molecular interactions but also in predicting molecular binding interfaces."

Reviewers' Comments:

Reviewer #2:

Remarks to the Author:

Thanks for the author's response! My concerns have been fully addressed. I stand for the acceptance of this work!

Reviewer #2 (Remarks to the Author):

Thanks for the author's response! My concerns have been fully addressed. I stand for the acceptance of this work!

Response:

We thank the reviewer for the time taken to review our work and the constructive feedback provided.